# REINFORCEMENT LEARNING WITH HUMAN FEEDBACK: LEARNING DYNAMIC CHOICES VIA PESSIMISM

## ABSTRACT

In this paper, we study offline Reinforcement Learning with Human Feedback (RLHF) where we aim to learn the human's underlying reward and the MDP's optimal policy from a set of trajectories induced by human choices. Existing RLHF practices often focus on the simplified bandit-feedback setting or when human preferences are myopic. However, how to learn optimal policy from non-myopic human choices in a dynamic environment remains underinvestigated. In this work, we focus on the Dynamic Discrete Choice (DDC) model that covers all these cases. DCC, rooted in econometrics and decision theory, is widely used to model a human decision-making process with forward-looking and bounded rationality. In this paper, we propose a Dynamic-Choice-Pessimistic-Policy-Optimization (DCPPO) method. The method involves a three-stage process: The first step is to estimate the human behavior policy and the state-action value function via maximum likelihood estimation (MLE); the second step recovers the human reward function via minimizing Bellman mean squared error using the learned value functions; the third step is to plug in the learned reward and invoke pessimistic value iteration for finding a near-optimal policy. With only single-policy coverage (i.e., optimal policy) of the dataset, we prove that the suboptimality of DCPPO *almost* matches the classical pessimistic offline RL algorithm in terms of suboptimality's dependency on distribution shift and dimension. To the best of our knowledge, this paper presents the first theoretical guarantees for off-policy offline RLHF with dynamic discrete choice model.

## 1 INTRODUCTION

*Reinforcement Learning with Human Feedback* (RLHF) is an area in machine learning research that incorporates human guidance or preference to learn an optimal policy. In recent years, RLHF has achieved significant success in large language models, clinical trials, auto-driving, robotics, etc. (Ouyang et al., 2022; Gao et al., 2022; Glaese et al., 2022; Hussein et al., 2017; Jain et al., 2013; Kupcsik et al., 2018; Menick et al., 2022; Nakano et al., 2021; Novoseller et al., 2020). In RLHF, the learner does not have direct access to the reward signal but instead can only observe a historical record of visited states and human-preferred actions. Then the reward is leveraged to learn the optimal policy by implementing algorithms such as soft actor-critic (Lee et al., 2021; Liang et al., 2022) or proximal policy optimization (Ouyang et al., 2022; Liang et al., 2022).

Despite its great success, existing RLHF practice often focuses on the simplified bandit feedback setting or when human preferences are myopic. However, how to learn optimal policy from non-myopic human choices in a dynamic environment remains underinvestigated. In this paper, we focus on *Dynamic Discrete Choice* (DDC) model. Such model has been extensively studied in econometrics literature (Rust, 1987; Hotz & Miller, 1993; Hotz et al., 1994; Aguirregabiria & Mira, 2002; Kalouptsidi et al., 2021; Bajari et al., 2015; Chernozhukov et al., 2022). In a DDC model, the agent make decisions under unobservable perturbation, i.e. $\pi_h(a_h \mid s_h) = \mathrm{argmax}_a\{Q_h(s_h, a) + \epsilon_h(a)\}$, where $\epsilon_h$ is an unobservable random noise and $Q_h$ is the agent's action value function. Specifically, our setting covers (i) trajectory-level feedback, in which the human preference is over prompt and full response, such as in LLM ; (ii) myopic humans (Zhang & Yu, 2013), in which human prones to choose the best action in current state; (iii) max entropy inverse RL (Ziebart et al., 2008; Zeng et al., 2022; Sharma et al., 2017), in which expert's choice actions to be more favorable

than actions not taken, which is a harder problem due to the non-myopic dynamical decision making process of the experts. We leave a detailed comparison in Appendix A.

Challenges for RLHF under dynamic choice model are three-folded: (i) The agent must first learn human behavior policies from the feedback data. (ii) The agent's behavior is related to the cumulative reward in a dynamic environment. Therefore, we need to recover the unobservable reward of the current step from estimated behavior policies. (iii) We face the challenge of insufficient dataset coverage and large state space.

With these coupled challenges, we ask the following question:

*Without access to the reward function, can one learn the optimal pessimistic policy from merely human choices under the dynamic choice model?*

**Our Results.** In this work, we propose the D̲ynamic-C̲hoice-P̲essimistic-P̲olicy-O̲ptimization (DCPPO) algorithm. By addressing challenges (i)-(iii), our contributions are three folds: (i) For learning behavior policies in large state spaces, we employ maximum likelihood estimation to estimate state/action value functions with function approximation. We establish estimation error bounds for general model class with low covering number. (ii) Leveraging the learned value functions, we minimize the Bellman mean squared error (BMSE) through regression. This allows us to recover the unobservable reward from the learned policy. Additionally, we demonstrate that the error of our estimated reward can be efficiently controlled by an uncertainty quantifier. (iii) To tackle the challenge of insufficient coverage, we follow *the principle of pessimism*, by incorporating a penalty into the value function during value iteration. We establish the suboptimality of our algorithm with high probability with only single-policy coverage.

Our result matches existing pessimistic offline RL algorithms in terms of suboptimality's dependence on distribution shift and dimension, even in the absence of an observable reward. To the best of our knowledge, our results offer the first theoretical guarantee for pessimistic RL under the human dynamic choice model.

## 1.1 RELATED WORK

**Reinforcement Learning with Human Preference.** In recent years RLHF and inverse reinforcement learning (IRL) has been widely applied to robotics, recommendation system, and large language model (Ouyang et al., 2022; Lindner et al., 2022; Menick et al., 2022; Jaques et al., 2020; Lee et al., 2021; Nakano et al., 2021). However, there are various ways to incorporate human preferences or expertise into the decision-making process of an agent. Shah et al. (2015); Ouyang et al. (2022); Saha & Krishnamurthy (2022) learn reward from pairwise comparison and ranking. Pacchiano et al. (2021) study pairwise comparison with function approximation in pairwise comparison. Zhu et al. (2023) study various cases of preference-based-comparison in contextual bandit problem with linear function approximation. Wang et al. (2018) study how to learn a uniformly better policy of an MDP from an offline dataset by learning the advantage function. However, they cannot guarantee the learned policy converges to the optimal policy. Moreover, previous works in RLHF and max entropy inverse RL corresponds to bandit case in our setting and can be easily covered. For a detailed comparison, check Appendix A.

**Dynamic Discrete Choice Model.** Dynamic Discrete Choice (DDC) model is a widely studied choice model in econometrics and is closely related to reward learning in IRL and RLHF. In the DDC model, the human agent is assumed to make decisions under the presence of Gumbel noise (Type I Extreme Error)(Aguirregabiria & Mira, 2002; Chernozhukov et al., 2022; Bajari et al., 2015; Kalouptsidi et al., 2021; Adusumilli & Eckardt, 2019), i.e. under bounded rationality, and the task is to infer the underlying utility. A method highly related to our work is the *conditional choice probability* (CCP) algorithm (Hotz & Miller, 1993; Arcidiacono & Ellickson, 2011; Bajari et al., 2015; Adusumilli & Eckardt, 2019), in which the learner first estimate choice probability from the dataset, and then recover the underlying value function from the estimated dynamic choices. However, most work in econometrics cares for asymptotic $\sqrt{n}$-convergence of estimated utility and does not study finite sample estimation error. Moreover, their methods requires sufficient coverage dataset, which is hard to satisfy. In recent years, there has been work combining the dynamic discrete choice model and IRL. Zeng et al. (2022) prove the equivalence between DDC estimation

problem and maximum likelihood IRL problem, and propose an online gradient method for reward estimation under ergodic dynamics assumption. Zeng et al. (2023) reformulate the reward estimation in the DDC model into a bilevel optimization and propose a model-based approach by assuming an environment simulator.

**Offline Reinforcement Learning and Pessimism.**    The idea of introducing pessimism for offline RL to deal with distribution shift has been studied in recent years (Jin et al., 2021; Uehara et al., 2021). Jin et al. (2021) show that pessimism is sufficient to eliminate spurious correlation and intrinsic uncertainty when doing value iteration. Uehara et al. (2021) show that with single-policy coverage, i.e. coverage over the optimal policy, pessimism is sufficient to guarantee a $\mathcal{O}(n^{-1/2})$ suboptimality. In this paper, we connect RLHF with offline RL and show our algorithm achieves pessimism by designing an uncertainty quantifier that can tackle error from estimating reward functions, which is crucial in pessimistic value iteration.

## 1.2    NOTATIONS AND PRELIMINARIES

For a positive-semidefinitematrix $A \in \mathbb{R}^{d \times d}$ and vector $x \in \mathbb{R}^d$, we use $\|x\|_A$ to denote $\sqrt{x^\top A x}$. For an arbitrary space $\mathcal{X}$, we use $\Delta(\mathcal{X})$ to denote the set of all probability distribution on $\mathcal{X}$. For two vectors $x, y \in \mathbb{R}^d$, we denote $x \cdot y = \sum_i^d x_i y_i$ as the inner product of $x, y$. We denote the set of all probability measures on $\mathcal{X}$ as $\Delta(\mathcal{X})$. We use $[n]$ to represent the set of integers from 0 to $n - 1$. For every set $\mathcal{M} \subset \mathcal{X}$ for metric space $\mathcal{X}$, we define its $\epsilon$-covering number with respect to norm $\| \cdot \|$ by $N(\mathcal{M}, \| \cdot \|, \epsilon)$. We define a finite horizon MDP model $M = (\mathcal{S}, \mathcal{A}, H, \{P_h\}_{h \in [H]}, \{r_h\}_{h \in [H]})$, $H$ is the horizon length, in each step $h \in [H]$, the agent starts from state $s_h$ in the state space $\mathcal{S}$, chooses an action $a_h \in \mathcal{A}$ with probability $\pi_h(a_h \mid s_h)$, receives a reward of $r_h(s_h, a_h)$ and transits to the next state $s'$ with probability $P_h(s' \mid s_h, a_h)$. Here $\mathcal{A}$ is a finite action set with $|\mathcal{A}|$ actions and $P_h(\cdot|s_h, a_h) \in \Delta(s_h, a_h)$ is the transition kernel condition on state action pair $(s, a)$. For convenience we assume that $r_h(s, a) \in [0, 1]$ for all $(s, a, h) \in \mathcal{S} \times \mathcal{A} \times [H]$. Without loss of generality, we assume that the initial state of each episode $s_0$ is fixed. Note that this will not add difficulty to our analysis. For any policy $\pi = \{\pi_h\}_{h \in [H]}$ the state value function is $V_h^\pi(s) = \mathbb{E}_\pi \left[ \sum_{t=h}^H r_t(s_t, a_t) \mid s_h = s \right]$, and the action value function is $Q_h^\pi(s, a) = \mathbb{E}_\pi \left[ \sum_{t=h}^H r_t(s_t, a_t) \mid s_h = s, a_h = a \right]$, here the expectation $\mathbb{E}_\pi$ is taken with respect to the randomness of the trajectory induced by $\pi$, i.e. is obtained by taking action $a_t \sim \pi_t(\cdot \mid s_t)$ and observing $s_{t+1} \sim P_h(\cdot \mid s_t, a_t)$. For any function $f : \mathcal{S} \to \mathbb{R}$, we define the transition operator $\mathbb{P}_h f(s, a) = \mathbb{E}[f(s_{h+1}) \mid s_h = s, a_h = a]$. We also define the Bellman equation for any policy $\pi$, $V_h^\pi(s) = \langle \pi_h(a \mid s), Q_h^{\pi_b}(s, a) \rangle, Q_h^\pi(s, a) = r_h(s, a) + \mathbb{P}_h V_{h+1}^\pi(s, a)$. For an MDP we denote its optimal policy as $\pi^*$, and define the performance metric for any policy $\pi$ as $\mathrm{SubOpt}(\pi) = V_1^{\pi^*} - V_1^\pi$.

## 2    PROBLEM FORMULATION

In this paper, we aim to learn from a dataset of human choices under dynamic discrete choice model. Suppose we are provided with dataset $\mathcal{D} = \{\mathcal{D}_h = \{s_h^i, a_h^i\}_{i \in [n]}\}_{h \in [H]}$, containing $n$ trajectories collected by observing a single human behavior in a dynamic discrete choice model. Our goal is to learn the optimal policy $\pi^*$ of the underlying MDP. We assume that the agent is bounded-rational and makes decisions according to the dynamic discrete choice model (Rust, 1987; Hotz & Miller, 1993; Chernozhukov et al., 2022; Zeng et al., 2023). In dynamic discrete choice model, the agent's policy has the following characterization (Rust, 1987; Aguirregabiria & Mira, 2002; Chernozhukov et al., 2022), which *deviates from optimal policy* due to bounded rationality:

$$\pi_{b,h}(a \mid s) = \frac{\exp(Q_h^{\pi_b, \gamma}(s, a))}{\sum_{a' \in \mathcal{A}} \exp(Q_h^{\pi_b, \gamma}(s, a'))}, \tag{1}$$

here $Q_h^{\pi_b, \gamma}(\cdot, \cdot)$ works as the solution of the discounted Bellman equation,

$$V_h^{\pi_b, \gamma}(s) = \langle \pi_{b,h}(a \mid s), Q_h^{\pi_b, \gamma}(s, a) \rangle, \qquad Q_h^{\pi_b, \gamma}(s, a) = r_h(s, a) + \gamma \cdot \mathbb{P}_h V_{h+1}^{\pi_b, \gamma}(s, a) \tag{2}$$

for all $(s, a) \in \mathcal{S} \times \mathcal{A}$. Note that equation 2 differs from the original Bellman equation due to the presence of $\gamma$, which is a discount factor in $[0, 1]$, and measures the myopia of the agent. The case

of $\gamma = 0$ corresponds to a *myopic* human agent. Such choice model comes from the perturbation of noises,

$$\pi_{b,h}(\cdot \mid s_h) = \mathrm{argmax}_{a \in \mathcal{A}} \left\{ r_h(s_h, a) + \epsilon_h(a) + \gamma \cdot \mathbb{P}_h V_{h+1}^{\pi_b, \gamma}(s_h, a) \right\},$$

where $\{\epsilon_h(a)\}_{a \in \mathcal{A}}$ are i.i.d Gumbel noises that is observed by the agent but not the learner, $\{V_h^{\gamma, \pi_b}\}_{h \in [H]}$ is the value function of the agent. Such model is widely used to model human decision. We also remark that the state value function defined in equation 2 corresponds to the *ex-ante* value function in econometric studies (Aguirregabiria & Mira, 2010; Arcidiacono & Ellickson, 2011; Bajari et al., 2015). When considering Gumbel noise as part of the reward, the value function may have a different form. However, such a difference does not add complexity to our analysis.

## 3   REWARD LEARNING FROM HUMAN DYNAMIC CHOICES

In this section, we present a general framework of an offline algorithm for learning the reward of the underlying MDP. Our algorithm consists of two steps: (i) The first step is to estimate the agent behavior policy from the pre-collected dataset $\mathcal{D}$ by maximum likelihood estimation (MLE). We recover the action value functions $\{Q_h^{\pi_b, \gamma}\}_{h \in [H]}$ from equation 1 and the state value functions $\{V_h^{\pi_b, \gamma}\}_{h \in [H]}$ from equation 2 using function approximation. In Section 3.1, we analyze the error of our estimation and prove that for any model class with a small covering number, the error from MLE estimation is of scale $\tilde{\mathcal{O}}(1/n)$ in dataset distribution. We also remark that our result does not need the dataset to be well-explored, which is implicitly assumed in previous works (Zhu et al., 2023; Chen et al., 2020). (ii) We recover the underlying reward from the model class by minimizing a penalized Bellman MSE with plugged-in value functions learned in step (i). In Section 3.2, we study linear model MDP as a concrete example. Theorem 3.5 shows that the error of estimated reward can be bounded by an elliptical potential term for all $(s, a) \in \mathcal{S} \times \mathcal{A}$ in both settings. First, we make the following assumption for function approximation.

**Assumption 3.1** (**Function Approximation Model Class**). *We assume the existence of a model class $\mathcal{M} = \{\mathcal{M}_h\}_{h \in [H]}$ containing functions $f : \mathcal{S} \times \mathcal{A} \to [0, H]$ for every $h \in [H]$, and is rich enough to capture $r_h$ and $Q_h$, i.e. $r_h \in \mathcal{M}_h$, $Q_h \in \mathcal{M}_h$. We also assume a positive penalty $\rho(\cdot)$ defined on $\mathcal{M}$.*

In practice, $\mathcal{M}_h$ can be a (pre-trained) neural network or a random forest. We now present our algorithm for reward learning in RLHF.

---

**Algorithm 1** DCPPO: Reward Learning for General Model Class

---

**Require:** Dataset $\left\{ \mathcal{D}_h = \{s_h^i, a_h^i\}_{i \in [n]} \right\}_{h \in [H]}$, constant $\lambda > 0$, penalty function $\rho(\cdot)$, parameter $\beta$.

1: **for** step $h = H, \ldots, 1$ **do**
2:     Set $\widehat{Q}_h = \mathrm{argmax}_{Q \in \mathcal{M}_h} \frac{1}{n} \sum_{i=1}^n Q(s_h^i, a_h^i) - \log \left( \sum_{a' \in \mathcal{A}} \exp(Q(s_h^i, a')) \right)$.
3:     Set $\widehat{\pi}_h(a_h \mid s_h) = \exp(\widehat{Q}_h(s_h, a_h)) / \sum_{a' \in \mathcal{A}} \exp(\widehat{Q}_h(s_h, a')$.
4:     Set $\widehat{V}_h(s_h) = \langle \widehat{Q}_h(s_h, \cdot), \widehat{\pi}_h(\cdot \mid s_h) \rangle_{\mathcal{A}}$.
5:     Set $\widehat{r}_h(s_h, a_h) = \mathrm{argmin}_{r \in \mathcal{M}_h} \left\{ \sum_{i=1}^n \left( r_h(s_h^i, a_h^i) + \gamma \cdot \widehat{V}_{h+1}(s_{h+1}^i) - \widehat{Q}_h(s_h^i, a_h^i) \right)^2 + \lambda \rho(r) \right\}$.
6: **end for**
7: **Output:** $\{\widehat{r}_h\}_{h \in [H]}$.

---

### 3.1   FIRST STEP: RECOVERING HUMAN POLICY AND HUMAN STATE-ACTION VALUES

For every step $h$, we use maximum liklihood estimaton (MLE) to estimate the behaviour policy $\pi_{b,h}$, corresponds to $Q_h^{\pi_b, \gamma}(s, a)$ in a general model class $\mathcal{M}_h$. For each step $h \in [H]$, we have the log-likelihood function

$$L_h(Q) = \frac{1}{n} \sum_{i=1}^n \log \left( \frac{\exp(Q(s_h^i, a_h^i))}{\sum_{a' \in \mathcal{A}} \exp(Q(s, a'))} \right) \tag{3}$$

for $Q \in \mathcal{M}_h$, and we estimate $Q_h$ by maximizing equation 3. Note that by Equation equation 1, adding a constant on $Q_h^{\pi_b,\gamma}$ will produce the same policy under dynamic discrete model, and thus the real behavior value function is unidentifiable in general. For identification, we have the following assumption.

**Assumption 3.2** (**Model Identification**). *We assume that there exists one $a_0 \in \mathcal{A}$, such that $Q(s, a_0) = 0$ for every $s \in \mathcal{S}$.*

Note that this assumption does not affect our further analysis. Other identifications include parameter constraint (Zhu et al., 2023) or utility constraints Bajari et al. (2015). We can ensure the estimation of the underlying policy and corresponding value function is accurate in the states the agent has encountered. Formally, we have the following theorem,

**Theorem 3.3** (**Value Functions Recovery from Choice Model**). *With Algorithm 1 , we have*

$$\mathbb{E}_{\mathcal{D}_h}\big[\|\widehat{Q}_h(s_h, \cdot) - Q_h^{\pi_b,\gamma}(s_h, \cdot)\|_1^2\big] \leq \mathcal{O}\bigg(\frac{H^2 e^{2H} \cdot |\mathcal{A}|^2 \cdot \log\big(H \cdot N(\mathcal{M}_h, \|\cdot\|_\infty, 1/n)/\delta\big)}{n}\bigg)$$

*hold for every $h \in [H]$ with probability at least $1 - \delta$. Here $\mathbb{E}_{\mathcal{D}_h}[\cdot]$ means the expectation is taken on collected dataset $\mathcal{D}_h$, i.e. the mean value taken with respect to $\{s_h^i\}_{i \in [n]}$.*

*Proof.* See Appendix B for details. $\qquad\square$

Theorem 3.3 shows that we can efficiently learn $\pi_{b,h}$ from the dataset under identification assumption. As a result, we can provably recover the value functions by definition in Equation 1.

## 3.2 Reward Learning from Dynamic Choices

As a concrete example, we study the instantiation of Algorithm 1 for the linear model class. We define the function class $\mathcal{M}_h = \{f(\cdot) = \phi(\cdot)^\top \theta : \mathcal{S} \times \mathcal{A} \to \mathbb{R}, \theta \in \Theta\}$ for $h \in [H]$, where $\phi \in \mathbb{R}^d$ is the feature defined on $\mathcal{S} \times \mathcal{A}$, $\Theta$ is a subset of $\mathbb{R}^d$ which parameterizes the model class, and $d > 0$ is the dimension of the feature. Corresponding to Assumption 3.2, We also assume that $\phi(s, a_0) = 0$ for every $s \in \mathcal{S}$. Note that this model class contains the reward $r_h$ and state action value function $Q_h$ in tabular MDP where $\phi(s, a)$ is the one-hot vector of $(s, a)$. The linear model class also contains linear MDP, which assumes both the transition $P(s_{h+1} \mid s_h, a_h)$ and the reward $r_h(s_h, a_h)$ are linear functions of feature $\phi(s_h, a_h)$ (Jin et al., 2020; Duan et al., 2020; Jin et al., 2021). In linear model case, our first step MLE in equation 3 turns into a logistic regression,

$$\widehat{\theta}_h = \mathrm{argmax}_{\theta \in \Theta} \frac{1}{n} \sum_{i=1}^n \phi(s_h^i, a_h^i) \cdot \theta - \log\bigg(\sum_{a' \in \mathcal{A}} \exp(\phi(s_h^i, a') \cdot \theta)\bigg), \qquad (4)$$

which can be efficiently solved by existing state-of-art optimization methods. We now have $\{\widehat{Q}_h\}_{h \in [H]}, \{\widehat{\pi}_h\}_{h \in [H]}$ and $\{\widehat{V}_h\}_{h \in [H]}$ in Algorithm 1 to be our estimations for $\{Q_h^{\pi_b,\gamma}\}_{h \in [H]}, \{\pi_{b,h}\}_{h \in [H]}$ and $\{V_h^{\pi_b,\gamma}\}_{h \in [H]}$. The second stage estimation in Line 5 of Algorithm 1 now turns into a ridge regression for the Bellman MSE, with $\rho(\phi \cdot w)$ being $\|w\|_2^2$,

$$\widehat{w}_h = \mathrm{argmin}_w \bigg\{ \sum_{i=1}^n \bigg(\phi(s_h^i, a_h^i) \cdot w + \gamma \cdot \widehat{V}_{h+1}(s_{h+1}^i) - \widehat{Q}_h(s_h^i, a_h^i)\bigg)^2 + \lambda \|w\|_2^2 \bigg\}. \qquad (5)$$

Note that equation 5 has a closed form solution,

$$\widehat{w}_h = (\Lambda_h + \lambda I)^{-1}\bigg(\sum_{i=1}^n \phi(s_h^i, a_h^i)\big(\widehat{Q}_h(s_h^i, a_h^i) - \gamma \cdot \widehat{V}_{h+1}(s_{h+1}^i)\big)\bigg) \qquad (6)$$

with $\Lambda_h = \sum_{i=1}^n \phi(s_h^i, a_h^i)\phi(s_h^i, a_h^i)^\top$, and we set $\widehat{r}(s_h, a_h) = \phi(s_h, a_h) \cdot \widehat{w}_h$. We also make the following assumption on the model class $\Theta$ and the feature function.

**Assumption 3.4** (**Regular Conditions**). *We assume that: (i) For all $\theta \in \Theta$, we have $\|\theta\|_2 \leq H\sqrt{d}$; for reward $r_h = \phi \cdot w_h$, we assume $\|w_h\|_2 \leq \sqrt{d}$. (ii) For all $(s_h, a_h) \in \mathcal{S} \times \mathcal{A}$, $\|\phi(s_h, a_h)\|_2 \leq 1$. (iii) For all $n > 0$, $\log N(\Theta, \|\cdot\|_\infty, 1/n) \leq c \cdot d \log n$ for some absolute constant $c$.*

We are now prepared to highlight our main result:

**Theorem 3.5** (**Reward Estimation for Linear Model MDP**). *With Assumption 3.1, 3.4, the estimation of our reward function holds with probability $1 - \delta$ for all $(s, a) \in \mathcal{S} \times \mathcal{A}$ and all $\lambda > 0$,*

$$|r_h(s, a) - \widehat{r}_h(s, a)| \leq \|\phi(s, a)\|_{(\Lambda_h + \lambda I)^{-1}} \cdot \mathcal{O}\left(\sqrt{\lambda d} + (1 + \gamma) \cdot He^H \cdot |\mathcal{A}| \cdot d\sqrt{\log\left(nH/\lambda\delta\right)}\right).$$

*Proof.* See Appendix C for details. □

Note that the error can be bounded by the product of two terms, the elliptical potential term $\|\phi(s, a)\|_{(\Lambda + \lambda \cdot I)^{-1}}$ and the norm of a self normalizing term of scale $O(He^H \cdot |\mathcal{A}| \cdot d\sqrt{\log(n/\delta)})$. Here the exponential dependency $\mathcal{O}(e^H|\mathcal{A}|)$ comes from estimating $Q_h^{\pi_b, \gamma}$ with logistic regression and also occurs in logistic bandit (Zhu et al., 2023; Fei et al., 2020). It remains an open question if this additional factor can be improved, and we leave it for future work.

**Remark 3.6.** *We remark that except for the exponential term in $H$, Theorem 3.5 almost matches the result when doing linear regression on an observable reward dataset, in which case error of estimation is of scale $\tilde{\mathcal{O}}(\|\phi(s, a)\|_{(\Lambda + \lambda I)^{-1}} \cdot dH)$ (Ding et al., 2021; Jin et al., 2021). When the human behavior policy has sufficient coverage, i.e. the minimal eigenvalue of $\mathbb{E}_{\pi_b}[\phi\phi^\top]$, $\sigma_{\min}(\mathbb{E}_{\pi_b}[\phi\phi^\top]) > c > 0$, we have $\|\phi(s, a)\|_{(\Lambda_h + \lambda I)^{-1}} = \mathcal{O}(n^{-1/2})$ holds for all $(s, a) \in \mathcal{S} \times \mathcal{A}$ (Duan et al., 2020) and $\|r_h - \widehat{r}_h\|_\infty = \mathcal{O}(n^{-1/2})$. However, even without strong assumptions such as sufficient coverage, we can still prove we can still achieve $\mathcal{O}(n^{-1/2})$ suboptimality with pessimistic value iteration.*

## 4 POLICY LEARNING FROM DYNAMIC CHOICES VIA PESSIMISTIC VALUE ITERATION

In this section, we describe the pessimistic value iteration algorithm, which minus a penalty function $\Gamma_h : \mathcal{S} \times \mathcal{A} \to \mathbb{R}$ from the value function when choosing the best action. Pessimism is achieved when $\Gamma_h$ is a *uncertainty quantifier* for our learned value functions $\{\tilde{V}_h\}_{h \in [H]}$, i.e.

$$\left|\left(\widehat{r}_h + \widetilde{\mathbb{P}}_h \widetilde{V}_{h+1}\right)(s, a) - \left(r_h + \mathbb{P}_h \widetilde{V}_{h+1}\right)(s, a)\right| \leq \Gamma_h(s, a) \text{ for all } (s, a) \in \mathcal{S} \times \mathcal{A} \quad (7)$$

with high probability. Then we use $\{\Gamma_h\}_{h \in [H]}$ as the penalty function for pessimistic planning, which leads to a conservative estimation of the value function. We formally describe our planning method in Algorithm 2. However, when doing pessimistic value iteration with $\{\widehat{r}_h\}_{h \in [H]}$ learned from human feedback, it is more difficult to design uncertainty quantifiers in equation 7, since the estimation error from reward learning is inherited in pessimistic planning. In Section 4.1, we propose an efficient uncertainty quantifier and prove that with pessimistic value iteration, Algorithm 2 can achieve a $\mathcal{O}(n^{-1/2})$ suboptimality gap even without any observable reward signal, which matches current standard results in pessimistic value iteration such as (Jin et al., 2021; Uehara & Sun, 2021; Uehara et al., 2021).

---

**Algorithm 2** DCPPO: Pessimistic Value iteration

**Require:** Surrogate reward $\{\widehat{r}_h(s_h, a_h)\}_{h \in [H]}$ learned in Algorithm 1, collected dataset $\{(s_h^i, a_h^i)\}_{i \in [n], h \in [H]}$, parameter $\beta$, penalty .

**Initialization:** Set $\widetilde{V}_{H+1}(s_{H+1}) = 0$.

1: **for** step $h = H, \ldots, 1$ **do**
2:   Set $\widetilde{\mathbb{P}}_h \widetilde{V}_{h+1}(s_h, a_h) = \text{argmin}_f \sum_{i \in [n]} \left(f(s_h^i, a_h^i) - \widetilde{V}_{h+1}(s_{h+1})\right)^2 + \lambda \cdot \rho(f)$.
3:   Construct $\Gamma_h(s_h, a_h)$ based on $\mathcal{D}$.
4:   Set $\widetilde{Q}_h(s_h, a_h) = \min\left\{\widehat{r}_h(s_h, a_h) + \widetilde{P}_h \widetilde{V}_{h+1}(s_h, a_h) - \Gamma_h(s_h, a_h), H - h + 1\right\}_+$.
5:   Set $\widetilde{\pi}_h(\cdot \mid s_h) = \text{argmax}\langle \widetilde{Q}_h(s_h, \cdot), \pi_h(\cdot \mid s_h)\rangle$.
6:   Set $\widetilde{V}_h(s_h) = \langle \tilde{Q}_h(s_h, \cdot), \tilde{\pi}_h(\cdot \mid s_h)\rangle_{\mathcal{A}}$.
7: **end for**
8: **Output:** $\{\widetilde{\pi}_h\}_{h \in [H]}$.

---

## 4.1 SUBOPTIMALITY GAP OF PESSIMITIC OPTIMAL POLICY

For linear model class defined in Section 3.2, we assume that we can capture the conditional expectation of value function in the next step with the known feature $\phi$. In formal words, we make the following assumption.

**Assumption 4.1 (Linear MDP).** *For the underlying MDP, we assume that for every $V_{h+1} : \mathcal{S} \to [0, H-h]$, there exists $u_h \in \mathbb{R}^d$ such that*

$$\mathbb{P}_h V_{h+1}(s, a) = \phi(s, a) \cdot u_h$$

*for all $(s, a) \in \mathcal{S} \times \mathcal{A}$. We also assume that $\|u_h\| \leq (H - h + 1) \cdot \sqrt{d}$ for all $h \in [H]$.*

Note that this assumption is directly satisfied by linear MDP class (Jin et al., 2021, 2020; Yang & Wang, 2019). For linear model MDP defined in Section 3.2, it suffices to have the parameter set $\Theta$ being closed under subtraction, i.e. if $x, y \in \Theta$ then $x - y \in \Theta$. Meanwhile, we construct $\Gamma_h$ in Algorithm 2 based on dataset $\mathcal{D}$ as

$$\Gamma_h(s, a) = \beta \cdot \left( \phi(s, a)^\top (\Lambda_h + \lambda I)^{-1} \phi(s, a) \right)^{1/2} \tag{8}$$

for every $h \in [H]$. Here that $\Lambda_h$ is defined in equation 6. To establish suboptimality for Algorithm 2, we assume that the trajectory induced by $\pi^*$ is "covered" by $\mathcal{D}$ sufficiently well.

**Assumption 4.2 (Single-Policy Coverage).** *Suppose there exists an absolute constant $c^\dagger > 0$ such that*

$$\Lambda_h \geq c^\dagger \cdot n \cdot \mathbb{E}_{\pi^*} \left[ \phi(s_h, a_h) \phi(s_h, a_h)^\top \right]$$

*holds with probability at least $1 - \delta/2$.*

We remark that Assumption 4.2 only assumes the human behavior policy can cover the optimal policy and is therefore weaker than assuming a well-explored dataset, or sufficient coverage e(Duan et al., 2020; Jin et al., 2021). With this assumption, we prove the following theorem.

**Theorem 4.3 (Suboptimality Gap for DCPPO).** *Suppose Assumption 3.2, 3.4, 4.1,4.2 holds. With $\lambda = 1$ and $\beta = \mathcal{O}(He^H \cdot |\mathcal{A}| \cdot d\sqrt{\log(nH/\delta)})$, we have (i) $\Gamma_h$ defined in equation 8 being uncertainty quantifiers, and (ii)*

$$\text{SubOpt} \left( \{\tilde{\pi}_h\}_{h \in [H]} \right) \leq c \cdot (1 + \gamma)|\mathcal{A}| d^{3/2} H^2 e^H n^{-1/2} \sqrt{\xi}$$

*holds for Algorithm 2 with probability at least $1 - \delta$ , here $\xi = \log(dHn/\delta)$. In particular, if $\text{rank}(\Sigma_h) \leq r$ at each step $h \in [H]$, then*

$$\text{SubOpt} \left( \{\tilde{\pi}_h\}_{h \in [H]} \right) \leq c \cdot (1 + \gamma)|\mathcal{A}| r^{1/2} dH^2 e^H n^{-1/2} \sqrt{\xi},$$

*here $\Sigma_h = \mathbb{E}_{\pi_b}[\phi(s_h, a_h)\phi(s_h, a_h)^\top]$.*

*Proof.* See Appendix D for detailed proof. $\qquad \square$

**Remark.** It is worth highlighting that Theorem 4.3 nearly matches the standard result for pessimistic offline RL with observable rewards in terms of the dependence on data size and distribution, up to a constant factor of $\mathcal{O}(|\mathcal{A}|e^H)$ (Jin et al., 2020; Uehara & Sun, 2021), where their suboptimality is of $\tilde{\mathcal{O}}(dH^2 n^{-1/2})$. Therefore, Algorithm 1 and 2 *almost* matches the suboptimality gap of standard pessimism planning with an observable reward, except for a $\mathcal{O}(e^H)$ factor inherited from reward estimation.

## 5 DCPPO FOR REPRODUCING KERNEL HILBERT SPACE

In this section, we assume the model class $\mathcal{M} = \{\mathcal{M}_h\}_{h \in [H]}$ are subsets of a Reproducing Kernel Hilbert Space (RKHS). For notations simplicity, we let $z = (s, a)$ denote the state-action pair and denote $\mathcal{Z} = \mathcal{S} \times \mathcal{A}$ for any $h \in [H]$. We view $\mathcal{Z}$ as a compact subset of $\mathbb{R}^d$ where the dimension $d$ is fixed. Let $\mathcal{H}$ be an RKHS of functions on $\mathcal{Z}$ with kernel function $K : \mathcal{Z} \times \mathcal{Z} \to \mathbb{R}$, inner product

$\langle \cdot, \cdot \rangle : \mathcal{H} \times \mathcal{H} \to \mathbb{R}$ and RKHS norm $\| \cdot \|_{\mathcal{H}} : \mathcal{H} \to \mathbb{R}$. By definition of RKHS, there exists a feature mapping $\phi : \mathcal{Z} \to \mathcal{H}$ such that $f(z) = \langle f, \phi(z) \rangle_{\mathcal{H}}$ for all $f \in \mathcal{H}$ and all $z \in \mathcal{Z}$. Also, the kernel function admits the feature representation $K(x, y) = \langle \phi(x), \phi(y) \rangle_{\mathcal{H}}$ for any $x, y \in \mathcal{H}$. We assume that the kernel function is uniformly bounded as $\sup_{z \in \mathcal{Z}} K(z, z) < \infty$. For notation simplicity, we assume that the discount factor $\gamma = 1$.

Let $\mathcal{L}^2(\mathcal{Z})$ be the space of square-integrable functions on $\mathcal{Z}$ and let $\langle \cdot, \cdot \rangle_{\mathcal{L}^2}$ be the inner product for $\mathcal{L}^2(\mathcal{Z})$. We define the Mercer operater $T_K : \mathcal{L}^2(\mathcal{Z}) \to \mathcal{L}^2(\mathcal{Z})$,

$$T_K f(z) = \int_{\mathcal{Z}} K(z, z') \cdot f(z') \, \mathrm{d}z', \quad \forall f \in \mathcal{L}^2(\mathcal{Z}). \tag{9}$$

In what follows, we assume the eigenvalue of the integral operator defined in 9 has a certain decay condition.

**Assumption 5.1** (Eigenvalue Decay of $\mathcal{H}$). *Let $\{\sigma_j\}_{j \geq 1}$ be the eigenvalues induced by the integral opretaor $T_K$ defined in Equation equation 9 and $\{\psi_j\}_{j \geq 1}$ be the associated eigenfunctions. We assume that $\{\sigma_j\}_{j \geq 1}$ satisfies one of the following conditions for some constant $\mu > 0$.*

   *(i) $\mu$-finite spectrum: $\sigma_j = 0$ for all $j > \mu$, where $\mu$ is a positive integer.*

   *(ii) $\mu$-exponential decay: there exists some constants $C_1, C_2 > 0, \tau \in [0, 1/2)$ and $C_\psi > 0$ such that $\sigma_j \leq C_1 \cdot \exp(-C_2 \cdot j^\mu)$ and $\sup_{z \in \mathcal{Z}} \sigma_j^\tau \cdot |\psi_j(z)| \leq C_\psi$ for all $j \geq 1$.*

   *(iii) $\mu$-polynomial decay: there exists some constants $C_1 > 0, \tau \in [0, 1/2)$ and $C_\psi > 0$ such that $\sigma_j \leq C_1 \cdot j^{-\mu}$ and $\sup_{z \in \mathcal{Z}} \sigma_j^\tau \cdot |\psi_j(z)| \leq C_\psi$ for all $j \geq 1$, where $\mu > 1$.*

For a detailed discussion of eigenvalue decay in RKHS, we refer the readers to Section 4.1 of Yang et al. (2020).

### 5.1 GURANTEE FOR RKHS

In RKHS case, our first step MLE in equation 3 turns into a kernel logistic regression,

$$\bar{Q}_h = \operatorname{argmin}_{Q \in \mathcal{H}} \frac{1}{n} \sum_{i=1}^{n} Q(s_h^i, a_h^i) - \log\left( \sum_{a' \in \mathcal{A}} \exp(Q(s, a')) \right). \tag{10}$$

Line 5 in Algorithm 1 now turns into a kernel ridge regression for the Bellman MSE, with $\rho(f)$ being $\|f\|_{\mathcal{H}}^2$,

$$\widehat{r}_h = \operatorname{argmin}_{r \in \mathcal{H}} \left\{ \sum_{i=1}^{n} \left( r(s_h^i, a_h^i) + \gamma \cdot \widehat{V}_{h+1}(s_{h+1}^i) - \widehat{Q}_h(s_h^i, a_h^i) \right)^2 + \lambda \|r\|_{\mathcal{H}}^2 \right\}. \tag{11}$$

Following Representer's Theorem (Steinwart & Christmann, 2008), we have the following closed form solution

$$\widehat{r}_h(z) = k_h(z)^\top (K_h + \lambda \cdot I)^{-1} y_h,$$

where we define the Gram matrix $K_h \in \mathbb{R}^{n \times n}$ and the function $k_h : \mathcal{Z} \to \mathbb{R}^n$ as

$$K_h = \left[ K\left(z_h^i, z_h^{i'}\right) \right]_{i,i' \in [n]} \in \mathbb{R}^{n \times n}, \quad k_h(z) = \left[ K\left(z_h^i, z\right) \right]_{i \in [n]}^\top \in \mathbb{R}^n, \tag{12}$$

and the entry of the response vector $y_h \in \mathbb{R}^n$ corresponding to $i \in [n]$ is

$$[y_h]_i = \widehat{Q}_h(s_h^i, a_h^i) - \gamma \cdot \widehat{V}_{h+1}\left(s_{h+1}^i\right).$$

Meanwhile, we also construct the uncertainty quantifier $\Gamma_h$ in Algorithm 2,

$$\Gamma_h(z) = \beta \cdot \lambda^{-1/2} \cdot \left( K(z, z) - k_h(z)^\top (K_h + \lambda I)^{-1} k_h(z) \right)^{1/2} \tag{13}$$

for all $z \in \mathcal{Z}$. Parallel to Assumption 4.1, we impose the following assumption for the kernel setting.

**Assumption 5.2.** *Let $R_r > 0$ be some fixed constant and we define function class $\mathcal{Q} = \{f \in \mathcal{H} : \|f\|_{\mathcal{H}} \leq HR_r\}$. We assume that $\mathbb{P}_h V_{h+1} \in \mathcal{Q}$ for any $V_{h+1} : \mathcal{S} \to [0, H]$. We also assume that $\|r\|_{\mathcal{H}} \leq R_r$ for some constant $R_r > 0$. We set the model class $\mathcal{M}_h = \mathcal{Q}$ for all $h \in [H]$.*

The above assumption states that the Bellman operator maps any bounded function into a bounded RKHS-norm ball, and holds for the special case of linear MDP Jin et al. (2021).

Besides the closeness assumption on the Bellman operator, we also define the maximal information gain (Srinivas et al., 2009) as a characterization of the complexity of $\mathcal{H}$ :

$$G(n, \lambda) = \sup \left\{ 1/2 \cdot \log \det \left( I + K_{\mathcal{C}}/\lambda \right) : \mathcal{C} \subset \mathcal{Z}, |\mathcal{C}| \leq n \right\} \tag{14}$$

Here $K_{\mathcal{C}}$ is the Gram matrix for the set $\mathcal{C}$, defined similarly as Equation equation 12.

We are now ready to present our results for RKHS setting. Theorem 5.3 establishes the concrete suboptimality of DCPPO under various eigenvalue decay conditions.

**Theorem 5.3** (**Suboptimality Gap for RKHS**). *Suppose that Assumption 5.1 holds. For $\mu$-polynomial decay, we further assume $\mu(1 - 2\tau) > 1$. For Algorithm 1 and 2, we set*

$$\lambda = \begin{cases} C \cdot \mu \cdot \log(n/\delta) & \mu\text{-finite spectrum,} \\ C \cdot \log(n/\delta)^{1+1/\mu} & \mu\text{-exponential decay,} \\ C \cdot (n/H)^{\frac{2}{\mu(1-2\tau)-1}} \cdot \log(n/\delta) & \mu\text{-polynomial decay,} \end{cases}$$

*and*

$$\beta = \begin{cases} C'' \cdot H \cdot \left\{ \sqrt{\lambda} R_r + d_{\text{eff}}^{sample} e^H |\mathcal{A}| \cdot \log(nR_r H/\delta)^{1/2+1/(2\mu)} \right\} & \mu\text{-finite spectrum,} \\ C'' \cdot H \cdot \left\{ \sqrt{\lambda} R_r + d_{\text{eff}}^{sample} e^H |\mathcal{A}| \cdot \log(nR_r H/\delta)^{1/2+1/(2\mu)} \right\} & \mu\text{-exponential decay,} \\ C'' \cdot H \cdot \left\{ \sqrt{\lambda} R_r + d_{\text{eff}}^{sample} e^H |\mathcal{A}| \cdot (nR_r)^{\kappa^*} \cdot \sqrt{\log(nR_r H/\delta)} \right\} & \mu\text{-polynomial decay.} \end{cases}$$

*Here $C > 0$ is an absolute constant that does not depend on $n$ or $H$. Then with probability at least $1 - \delta$, it holds that (i) $\Gamma_h$ set in equation 13 being uncertainty quantifiers, and (ii)*

$$\text{SubOpt}(\{\widetilde{\pi}_h\}_{h\in[H]}) \leq \begin{cases} C' \cdot \tilde{d} \cdot H e^H |\mathcal{A}| \sqrt{\mu \cdot \log(nR_r H/\delta)} \} & \mu\text{-finite spectrum,} \\ C' \cdot \tilde{d} \cdot H e^H |\mathcal{A}| \cdot \sqrt{(\log(nR_r H)/\delta)^{1+1/\mu}} & \mu\text{-exponential decay,} \\ C' \cdot \tilde{d} \cdot H e^H |\mathcal{A}| \cdot (nR_r)^{\kappa^*} \cdot \sqrt{\log(nR_r H/\delta)} & \mu\text{-polynomial decay.} \end{cases}$$

*Here $C, C', C''$ are absolute constants irrelevant to $n$ and $H$ and $\tilde{d} = d_{\text{eff}}^{pop} \cdot d_{\text{eff}}^{sample}$, $\kappa^* = \frac{d+1}{2(\mu+d)} + \frac{1}{\mu(1-2\tau)-1}$. Here $d_{\text{eff}}^{pop}$ is the population effective dimension, which measures the "coverage" of the human behavior $\pi_b$ for the optimal policy $\pi^*$.*

*Proof.* See Appendix E.2 for detailed proof. $\square$

For simplicity of notation, we delay the formal definition of $d_{\text{eff}}^{\text{sample}}$ and $d_{\text{eff}}^{\text{pop}}$ to the appendix. If the behavior policy is close to the optimal policy and the RKHS satisfies Assumption 5.1, $d_{\text{eff}}^{\text{pop}} = \mathcal{O}(H^{3/2}n^{-1/2})$ and $d_{\text{eff}}^{\text{sample}}$ remains in constant level. In this case suboptimality is of order $\mathcal{O}(n^{-1/2})$ for $\mu$-finite spectrum and $\mu$-exponential decay, while for $\mu$-polynomial decay we obtain a rate of $\mathcal{O}(n^{\kappa^* -1/2})$. This also matches the results in standard pessimistic planning under RKHS case (Jin et al., 2021), where the reward is observable.

## 6 CONCLUSION

In this paper, we have developed a provably efficient online algorithm, Dynamic-Choice-Pessimistic-Policy-Optimization (DCPPO) for RLHF under dynamic discrete choice model. By maximizing log-likelihood function of the Q-value function and minimizing mean squared Bellman error for the reward, our algorithm learns the unobservable reward, and the optimal policy following the principle of pessimism. We prove that our algorithm is efficient in sample complexity for linear model MDP and RKHS model class. To the best of our knowledge, this is the first provably efficient algorithm for offline RLHF under the dynamic discrete choice model.

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

# A COMPARISON TO EXISTING RLHF METHODS AND MAX ENTROPY INVERSE RL

**Existing methods in RLHF.** We give a detailed comparison between our setting and existing RLHF works such as Ouyang et al. (2022); Liang et al. (2022); Stiennon et al. (2022); Christiano et al. (2023); Lee et al. (2021). In those works, the agent interacts with the environment base on trajectory-level feedbacks: (i) a model generates several trajectories $\{\{\sigma_i^j\}_{i\in[2]}\}_{j\in[n]}$ given i.i.d. prompts $\{p^j\}_{j\in[n]}$; (ii) the human scorer ranks her preference $y^j \in \{1, 2\}$ by a probability of

$$\mathbb{P}(\sigma_1^j \succ \sigma_2^j \mid p^j) = \mathbb{P}(y^j = 1) = \frac{\exp(r(p^j, \sigma_1^j))}{\exp(r(p^j, \sigma_1^j)) + \exp(r(p^j, \sigma_2^j))}. \tag{15}$$

The algorithm then (i) collects the human feedback dataset $\mathcal{D} = \{(\sigma_1^j, \sigma_2^j, y^j)\}_{j\in[n]}$ and learns the reward by MLE:

$$\widehat{r} \in \arg\min_{r\in\mathcal{F}}\ell_{\mathcal{D}}(r),$$

$$\text{where } \ell_{\mathcal{D}}(r) = -\sum_{i=1}^{n} \log\left(\frac{1\left(y^j = 1\right) \cdot \exp(r(\sigma_1^j))}{\exp(r(\sigma_1^j)) + \exp(r(\sigma_2^j))} + \frac{1\left(y^j = 0\right) \cdot \exp(r(\sigma_2^j))}{\exp(r(\sigma_1^j)) + \exp(r(\sigma_2^j))}\right)$$

where $\mathcal{F}$ is a function class, e.g. a neural network; (ii) the algorithm uses reinforcement learning methods such as proximal policy optimization or soft actor-critic to learn the optimal policy: $\pi(\sigma \mid p) = \max_\sigma \widehat{r}(p, \sigma)$.

We would like to point out these papers consider *RLHF in static case* since they consider why the human prefers a whole trajectory over others. Due to the trajectories i.i.d. generated by an underlying model, the agent and the algorithm are *myopic*, since they only take the reward of the instant choice into account. In this paper, we consider dynamic cases – why the human agent iteratively makes choices in different states, which is more challenging due to the dynamic nature of MDP transition. Specifically, current trajectory-based RLHF settings in large language models such as Ouyang et al. (2022); Liang et al. (2022); Stiennon et al. (2022); Christiano et al. (2023) can be taken as a special case of DDC model: by taking $H = 1$ and $s_1$ being the concatenation of input prompt and multiple trajectories generated by the pre-trained model, i.e. $s_1 = (p, \{\sigma_i\}_{i\in[2]})$, and the action set $\mathcal{A} = \{1, 2\}$, the human choice probability under DDC is

$$\mathbb{P}(a = 1 \mid s_1) = \frac{\exp(r(s_1, \sigma_1))}{\exp(r(s_1, \sigma_1)) + \exp(r(p, \sigma_2))},$$

which exactly recovers equation 15. Note that such a setting lies in the *contextual bandit* case, in which action selections do not impact future state transitions, while our algorithm is more general can handle broader cases with non-myopic agents.

**Existing methods in inverse RL.** Another concept closely related to our paper is *inverse reinforcement learning*. In entropy-based inverse RL work such as Wulfmeier et al. (2016); Ziebart et al. (2008); Zhou et al. (2020), the reward is unknown and we can only observe $\{\sigma^j\}_{j\in[n]}$, a set of trajectories generated by an agent, where $\sigma_j = \{(x_k^j)\}_{k\in[K]}$, with $x_k$ being the $k$-th context of the trajectory that could either be a single state or a state-action pair. The agent is assumed to be attempting to optimize $\sum_{k=1}^{K} c_k(x_k)$, where $c$ acts as the reward function of the agent. Following the principle of maximum entropy, the algorithm assumes that plans with higher rewards are exponentially more preferred, i.e. the agent chooses trajectory $\sigma$ with probability

$$\mathbb{P}(\sigma) = \frac{\exp\left(\sum_{k=1}^{K} c_k(x_k)\right)}{\sum_{\sigma'\in\mathcal{T}} \exp\left(\sum_{k=1}^{K} c_h(x_k')\right)},$$

here $\mathcal{T}$ represents all possible trajectories. The algorithm then recover the underlying reward $\{c_h\}_{h\in[H]}$ by the following optimization:

$$\{c_j\}_{k\in[K]} = \arg\max_{c_k\in\mathcal{F}, k\in[K]} \sum_{j\in[n]} \log\mathbb{P}(\sigma^j),$$

here $\mathcal{F}$ can either be a linear function class (Ziebart et al., 2008) or a deep neural network (Wulfmeier et al., 2016). We claim that such a setting is covered by our model. Specifically, set $H = 1$, and let $\mathcal{A} = \mathcal{T}$, i.e. the agent makes choices among all possible trajectories. For a trajectory $\sigma = \{x_k\}_{k \in [K]}$, set the reward $r_h(\sigma) = \sum_{k \in [K]} c(x_k)$. Then the human choice probability under DDC is

$$\mathbb{P}(a = \sigma) = \frac{\exp(r(a))}{\sum_{a' \in \mathcal{A}} \exp(r(a'))} = \frac{\exp\left(\sum_{k \in [K]} c_k(x_k)\right)}{\sum_{\sigma' \in \mathcal{T}} \exp(\sum_{k \in [K]} c_k(x'_k))},$$

which recovers the result in max entropy IRL. Moreover, we claim that Algorithm 1 is more general and realistic than the classical max entropy inverse RL, since we can handle the non-bandit case in which human makes preferences on single state-action pairs for each step $h \in [H]$, instead of only the whole trajectory set.

## B  PROOF FOR THEOREM 3.3

Theorem 3.3 can be regarded as an MLE guarantee for dataset distribution. Our proof for Theorem 3.3 lies in two steps: (i) We prove an MLE guarantee in population distribution, i.e. when $s_h$ is sampled by the behavior policy $\pi_{b,h}$, the estimation error can be bounded in expectation; (ii) With a concentration approach, we transfer the expectation bound to a bound on a realized dataset. First, for MLE with an identifiable $Q_h^{\pi_b, \gamma} \in \mathcal{M}_h$, we have the following guarantee:

**Lemma B.1** (MLE distribution bound). *For $\widehat{Q}_h$ estimated by equation 3, we have*

$$\mathbb{E}_{s_h \sim \pi_b}\left[\|\widehat{\pi}_h(\cdot \mid s_h) - \pi_{b,h}(\cdot \mid s_h)\|_1^2\right] \leq c \cdot \frac{\log\left(H \cdot N(\mathcal{M}, \|\cdot\|_\infty, 1/n)/\delta\right)}{n}$$

*with probability at least $1 - \delta$. Here $c, c' > 0$ are two absolute constants, and $\delta$ measures the confidence in the estimation.*

*Proof.* For all $h \in [H]$, define

$$\Pi_h = \left\{\pi_Q(a \mid s) = \exp(Q(s,a))/\sum_{a' \in \mathcal{A}} \exp(Q(s,a')) \text{ for some } Q \in \mathcal{M}_h\right\}.$$

Let $\mathcal{N}_{[]}(\Pi_h, \|\cdot\|_\infty, 1/n)$ be the smallest $1/n$-upper bracketing of $\Pi_h$. And $|\mathcal{N}_{[]}(\Pi_h, \|\cdot\|_\infty, 1/n)| = N_{[]}(\Pi_h, \|\cdot\|_\infty, 1/n)$, where $N_{[]}(\Pi_h, \|\cdot\|_\infty, 1/n)$ is the bracketing number of $\Pi_h$. First, we prove that

$$\mathbb{E}_{s_h \sim \pi_b}\left[\|\widehat{\pi}_h(\cdot \mid s_h) - \pi_{b,h}(\cdot \mid s_h)\|_1^2\right] \leq \mathcal{O}\left(\frac{\log\left(H \cdot N_{[]}(\Pi_h, \|\cdot\|_\infty, 1/n)/\delta\right)}{n}\right)$$

with probability at least $1 - \delta$. By MLE guarantee, we have

$$\frac{1}{n}\sum_{i=1}^n \log\left(\frac{\widehat{\pi}_h(a_h^t \mid s_h^t)}{\pi_{b,h}(a_h^t \mid s_h^t)}\right) \geq 0,$$

by Markov's inequality and Boole's inequality, it holds with probability at least $1 - \delta$ that for all $\bar{\pi} \in \mathcal{N}_{[]}(\Pi, \|\cdot\|_\infty, 1/n)$, we have

$$\sum_{i=1}^n \frac{1}{2}\log\left(\frac{\bar{\pi}(a_h^t \mid s_h^t)}{\pi_{b,h}(a_h^t \mid s_h^t)}\right) \leq n\log\left(\mathbb{E}_{\pi_b}\left[\exp\left(\frac{1}{2}\log\left(\frac{\bar{\pi}(\cdot \mid \cdot)}{\pi_{b,h}(\cdot \mid \cdot)}\right)\right)\right]\right) + \log\left(\frac{N_{[]}(\Pi, \|\cdot\|_\infty, 1/n)}{\delta}\right),$$

specify $\bar{\pi}$ to be the upper bracket of $\widehat{\pi}_h$, we have

$$0 \leq n\log\left(\mathbb{E}_{\pi_b}\left[\exp\left(\frac{1}{2}\log\left(\frac{\bar{\pi}(\cdot \mid \cdot)}{\pi_{b,h}(\cdot \mid \cdot)}\right)\right)\right]\right) + \log\left(\frac{N_{[]}(\Pi, \|\cdot\|_\infty, 1/n)}{\delta}\right)$$

$$\leq n \cdot \log\left(\mathbb{E}_{\pi_b}\left[\sqrt{\frac{\bar{\pi}(\cdot \mid \cdot)}{\pi_{b,h}(\cdot \mid \cdot)}}\right]\right) + \log\left(\frac{N_{[]}(\Pi, \|\cdot\|_\infty, 1/n)}{\delta}\right)$$

$$= n \cdot \log\left(\mathbb{E}_{s_h \sim \pi_b}\left[\sum_{a \in \mathcal{A}} \sqrt{\bar{\pi}(a \mid s_h) \cdot \pi_{b,h}(a \mid s_h)}\right]\right) + \log\left(\frac{N_{[]}(\Pi, \|\cdot\|_\infty, 1/n)}{\delta}\right),$$

Here $\mathbb{E}_{s_h \sim \pi_b}$ means $s_h$ is simulated by the policy $\pi_b$. Utilizing the $\log x \leq x - 1$, we have

$$1 - \mathbb{E}_{\pi_b}\left[\sum_{a \in \mathcal{A}} \sqrt{\bar{\pi}(a \mid s_h) \cdot \pi_{b,h}(a \mid s_h)}\right] \leq \frac{1}{n}\log\left(\frac{N_{[]}(\Pi, \|\cdot\|_\infty, 1/n)}{\delta}\right).$$

Therefore we can bound the Hellinger distance between $\pi_{b,h}$ and $\bar{\pi}$,

$$h(\bar{\pi}, \pi_{b,h}) = \mathbb{E}_{s_h \sim \pi_b}\left[\sum_{a \in \mathcal{A}}\left(\bar{\pi}(a \mid s_h)^{1/2} - \pi_{b,h}(a \mid s_h)^{1/2}\right)^2\right] \tag{16}$$

$$\leq 2\left(1 - \sum_{a \in \mathcal{A}}\sqrt{\bar{\pi}(a \mid s_h) \cdot \pi_{b,h}(a \mid s_h)}\right) + \frac{1}{n} \tag{17}$$

$$\leq \frac{2}{n}\log\left(\frac{N_{[]}(\Pi, \|\cdot\|_\infty, 1/n)}{\delta}\right) + \frac{1}{n}, \tag{18}$$

here the second inequality comes from the fact that $\bar{\pi}$ is a upper bracketing of $\Pi$. Moreover, it is easy to verify that

$$\mathbb{E}_{s_h \sim \pi_b}\left[\sum_{a \in \mathcal{A}}((\bar{\pi}(a \mid s_h)^{1/2} + \pi_{b,h}(a \mid s_h)^{1/2}))^2\right] \leq 2\mathbb{E}_{s_h \sim \pi_b}\left[\sum_{a \in \mathcal{A}}(\bar{\pi}(a \mid s_h) + \pi_{b,h}(a \mid s_h))\right] \tag{19}$$

$$\leq \frac{2}{n} + 4, \tag{20}$$

where the second inequality comes from the fact that $\bar{\pi}$ is the $1/n$-upper bracket of a probability distribution. Combining the equation 16 and equation 19, by Cauchy-Schwarz inequality, we have

$$\mathbb{E}_{s_h \sim \pi_b}\left[\|\bar{\pi}(\cdot \mid s_h) - \pi_{b,h}(\cdot \mid s_h)\|_1^2\right] \leq \frac{15}{n} \cdot \log\left(\frac{N_{[]}(\Pi, \|\cdot\|_\infty, 1/n)}{\delta}\right).$$

Meanwhile,

$$\|\bar{\pi}(\cdot \mid s_h) - \pi_{b,h}(\cdot \mid s_h)\|_1^2 - \|\hat{\pi}_h(\cdot \mid s_h) - \pi_{b,h}(\cdot \mid s_h)\|_1^2$$
$$\leq \left(\sum_{a \in \mathcal{A}}|\bar{\pi}(\cdot \mid s_h) - \pi_{b,h}(\cdot \mid s_h)| + \sum_{a \in \mathcal{A}}|\hat{\pi}_h(\cdot \mid s_h) - \pi_{b,h}(\cdot \mid s_h)|\right)$$
$$\cdot \left(\sum_{a \in \mathcal{A}}|\bar{\pi}(\cdot \mid s_h) - \pi_{b,h}(\cdot \mid s_h)| - \sum_{a \in \mathcal{A}}|\hat{\pi}_h(\cdot \mid s_h) - \pi_{b,h}(\cdot \mid s_h)|\right)$$
$$\leq (4 + \frac{1}{n}) \cdot \frac{1}{n},$$

therefore we have

$$\mathbb{E}_{s_h \sim \pi_b}\left[\|\hat{\pi}_h(\cdot \mid s_h) - \pi_{b,h}(\cdot \mid s_h)\|_1^2\right] \leq \frac{20}{n} \cdot \log\left(\frac{N_{[]}(\Pi_h, \|\cdot\|_\infty, 1/n)}{\delta}\right).$$

Next, we bound $N_{[]}(\Pi_h, \|\cdot\|_\infty, 1/n)$ by $N(\mathcal{M}_h, \|\cdot\|_\infty, 1/4n)$. For all $h \in [H]$, recall the definition

$$\Pi_h = \left\{\pi_Q(a \mid s) = \exp(Q(s,a))/\sum_{a' \in \mathcal{A}}\exp(Q(s,a')) \text{ for some } Q \in \mathcal{M}_h\right\},$$

it is easy to check that

$$|\pi_Q(a \mid s) - \pi_{Q'}(a \mid s)| \leq 2 \cdot \|Q - Q'\|_\infty, \forall(s,a) \in \mathcal{S} \times \mathcal{A}.$$

Recall that $N(\mathcal{M}_h, \|\cdot\|_\infty, 1/n)$ is the covering number for model class $\mathcal{M}_h$. Using Lemma F.3, we have

$$N_{[]}(\Pi_h, \|\cdot\|_\infty, 1/n) \leq N(\mathcal{M}_h, \|\cdot\|_\infty, 1/4n) \tag{21}$$

always hold for all $h \in [H]$. Therefore we have

$$\mathbb{E}_{s_h \sim \pi_b}\left[\|\hat{\pi}_h(\cdot \mid s_h) - \pi_{b,h}(\cdot \mid s_h)\|_1^2\right] \leq \mathcal{O}\left(\frac{\log(H \cdot N(\mathcal{M}_h, \|\cdot\|_\infty, 1/n)/\delta)}{n}\right)$$

holds for $h \in [H]$ with probability $1 - \delta/H$. Taking union bound on $h \in [H]$ and we conclude the proof for Lemma B.1. $\qquad\square$

## B.1 PROOF FOR THEOREM 3.3

From Lemma B.1, we have the following generalization bound: with probability $1 - \delta$,

$$\mathbb{E}_{s_h \sim \pi_b}\left[\|\widehat{\pi}_h(\cdot \mid s_h) - \pi_{b,h}(\cdot \mid s_h)\|_1^2\right] \leq \mathcal{O}\left(\frac{\log(H \cdot N(\mathcal{M}_h, \|\cdot\|_\infty, 1/n)/\delta)}{n}\right)$$

for all $h \in [H]$. We now condition on this event. Letting

$$A(\widehat{\pi}_h) := \left|\mathbb{E}_{s_h \sim \pi_b}\left[\|\widehat{\pi}_h(\cdot \mid s_h) - \pi_{b,h}(\cdot \mid s_h)\|_1^2\right] - \mathbb{E}_{\mathcal{D}_h}\left[\|\widehat{\pi}_h(\cdot \mid s_h) - \pi_{b,h}(\cdot \mid s_h)\|_1^2\right]\right|.$$

With probability $1 - \delta$, from Bernstein's inequality, we also have

$$A(\widehat{\pi}_h) \leq \mathcal{O}\left(\frac{\log(H/\delta)}{n} + \sqrt{\frac{\text{Var}_{s_h \sim \pi_b}[\|\widehat{\pi}_h(\cdot \mid s_h) - \pi_{b,h}(\cdot \mid s_h)\|_1^2]\log(H/\delta)}{n}}\right)$$

$$\leq \mathcal{O}\left(\frac{\log(H/\delta)}{n} + \sqrt{\frac{\mathbb{E}_{s_h \sim \pi_b}[\|\widehat{\pi}_h(\cdot \mid s_h) - \pi_{b,h}(\cdot \mid s_h)\|_1^2]\log(H/\delta)}{n}}\right)$$

$$\leq \mathcal{O}\left(\frac{\log(H \cdot N(\mathcal{M}_h, \|\cdot\|_\infty, 1/n)/\delta)}{n}\right).$$

holds for all $h \in [H]$ with probability at least $1 - \delta$, and therefore we have

$$\mathbb{E}_{\mathcal{D}_h}\left[\|\widehat{\pi}_h(\cdot \mid s_h) - \pi_{b,h}(\cdot \mid s_h)\|_1^2\right] \leq \mathcal{O}\left(\frac{\log(H \cdot N(\mathcal{M}_h, \|\cdot\|_\infty, 1/n)/\delta)}{n}\right),$$

i.e. the error of estimating $\pi_{b,h}$ decreases in scale $\tilde{O}(1/n)$ on the dataset. Recall that

$$\widehat{\pi}_h(a \mid s) = \frac{\exp(\widehat{Q}_h(s, a))}{\sum_{a' \in \mathcal{A}} \exp(\widehat{Q}_h(s_h, a'))}$$

and

$$\pi_{b,h}(a \mid s) = \frac{\exp(Q_h^{\pi_b, \gamma}(s, a))}{\sum_{a' \in \mathcal{A}} \exp(Q_h^{\pi_b, \gamma}(s, a'))}.$$

Also we have $\widehat{Q}_h(s, a_0) = Q_h^{\pi_b, \gamma}(s, a_0) = 0$ and $Q_h^{\pi_b, \gamma} \in [0, H]$ by Assumption 3.2 and definition of $\mathcal{M}_h$. Therefore, we have

$$\left|\widehat{Q}_h(s, a) - Q_h^{\pi_b, \gamma}(s, a)\right| = \left|\log\left(\frac{\widehat{\pi}_h(a \mid s)}{\pi_{b,h}(a \mid s)}\right) - \log\left(\frac{\widehat{\pi}_h(a_0 \mid s)}{\pi_{b,h}(a_0 \mid s)}\right)\right|.$$

Utilizing $\ln(x/y) \leq x/y - 1$ for $x, y > 0$, and $\pi_h(a \mid s) \in [e^{-H}, 1]$, we have

$$\left|\widehat{Q}_h(s, a) - Q_h^{\pi_b, \gamma}(s, a)\right| \leq e^H \cdot \left(|\pi_{b,h}(a \mid s) - \widehat{\pi}_h(a \mid s)| + |\pi_{b,h}(a_0 \mid s) - \widehat{\pi}_h(a_0 \mid s)|\right),$$

and by taking summation over $a \in \mathcal{A}$, we have

$$\mathbb{E}_{s_h \in \mathcal{D}_h}\left[\|\widehat{Q}_h(s_h, \cdot) - Q_h^{\pi_b, \gamma}(s_h, \cdot)\|_1^2\right] \leq c' \cdot \frac{H^2 e^{2H} \cdot |\mathcal{A}|^2 \cdot \log\left(H \cdot N(\mathcal{M}, \|\cdot\|_\infty, 1/n)/\delta\right)}{n},$$

and we complete our proof.

## C PROOF FOR THEOREM 3.5

Recall that in equation 6, we have

$$\widehat{w}_h = (\Lambda_h + \lambda I)^{-1}\left(\sum_{i=1}^n \phi(s_h^i, a_h^i)\left(\widehat{Q}_h(s_h^i, a_h^i) - \gamma \cdot \widehat{V}_{h+1}(s_{h+1}^i)\right)\right)$$

where

$$\Lambda_h = \sum_{i=1}^n \phi(s_h^i, a_h^i)\phi(s_h^i, a_h^i)^\top.$$

By Assumption 3.1, there exists $w_h \in \mathbb{R}^d$ such that $r_h(s,a) = \phi(s,a) \cdot w_h$. By our construction for $\widehat{r}_h$ in Algorithm 1, we therefore have

$$|r_h(s,a) - \widehat{r}_h(s,a)| = |\phi(s,a)(w_h - \widehat{w}_h)|$$

$$= \left| \phi(s,a)\big(\Lambda_h + \lambda I\big)^{-1}\left( \lambda \cdot w_h + \sum_{i=1}^{n} \phi(s_h^i, a_h^i)\big(\widehat{Q}_h(s_h^i, a_h^i) - \gamma \cdot \widehat{V}_{h+1}(s_{h+1}^i) - r_h(s_h^i, a_h^i)\big) \right) \right|$$

$$\leq \underbrace{\lambda \cdot |\phi(s,a)\big(\Lambda_h + \lambda I\big)^{-1} w_h|}_{\text{(i)}}$$

$$+ \underbrace{\left| \phi(s,a)\big(\Lambda_h + \lambda I\big)^{-1}\left( \sum_{i=1}^{n} \big(\widehat{Q}_h(s_h^i, a_h^i) - \gamma \cdot \widehat{V}_h(s_{h+1}^i) - r_h(s_h^i, a_h^i)\big) \right) \right|}_{\text{(ii)}},$$

For (i), we have

$$\text{(i)} \leq \lambda \cdot \|\phi(s,a)\|_{(\Lambda_h + \lambda I)^{-1}} \cdot \|w_h\|_{(\Lambda_h + \lambda I)^{-1}},$$

by Cauchy-Schwarz inequality and by $\Lambda_h$ being semi-positive definite and $\|w_h\|_2 \leq \sqrt{d}$, we have

$$\text{(i)} \leq \lambda \cdot \sqrt{d/\lambda} \cdot \|\phi(s,a)\|_{(\Lambda_h + \lambda I)^{-1}} = \sqrt{\lambda d} \cdot \|\phi(s,a)\|_{(\Lambda_h + \lambda I)^{-1}}, \tag{22}$$

and

$$\text{(ii)} \leq \|\phi(s,a)\|_{(\Lambda_h + \lambda I)^{-1}} \cdot \underbrace{\left\| \sum_{i=1}^{n} \phi(s_h^i, a_h^i)\big(\widehat{Q}_h(s_h^i, a_h^i) - \gamma \cdot \widehat{V}_{h+1}(s_{h+1}^i) - r_h(s_h^i, a_h^i)\big) \right\|_{(\Lambda_h + \lambda I)^{-1}}}_{\text{(iii)}}.$$

Recall that in equation 2, we have the following Bellman equation hold for all $(s_h, a_h) \in \mathcal{S} \times \mathcal{A}$,

$$r_h(s_h, a_h) + \gamma \cdot \mathbb{P}_h V_{h+1}^{\pi_b, \gamma}(s_h, a_h) = Q_h^{\pi_b, \gamma}(s_h, a_h),$$

substitute this into (iii), and we have

$$\text{(iii)} = \left\| \sum_{i=1}^{n} \phi(s_h^i, a_h^i)\left( \big(\widehat{Q}_h(s_h^i, a_h^i) - Q_h^{\pi_b, \gamma}(s_h^i, a_h^i)\big) - \gamma \cdot \big(\widehat{V}_{h+1}(s_{h+1}^i) - \mathbb{P}_h V_{h+1}^{\pi_b, \gamma}(s_{h+1}^i)\big) \right) \right\|_{(\Lambda_h + \lambda I)^{-1}}$$

$$\leq \underbrace{\left\| \sum_{i=1}^{n} \phi(s_h^i, a_h^i)\left( \big(\widehat{Q}_h(s_h^i, a_h^i) - Q_h^{\pi_b, \gamma}(s_h^i, a_h^i)\big) \right) \right\|_{(\Lambda_h + \lambda I)^{-1}}}_{\text{(iv)}}$$

$$+ \gamma \cdot \underbrace{\left\| \sum_{i=1}^{n} \phi(s_h^i, a_h^i)\left( \big(\widehat{V}_{h+1}(s_{h+1}^i) - V_{h+1}^{\pi_b, \gamma}(s_{h+1}^i)\big) \right) \right\|_{(\Lambda_h + \lambda I)^{-1}}}_{\text{(v)}}$$

$$+ \gamma \cdot \underbrace{\left\| \sum_{i=1}^{n} \phi(s_h^i, a_h^i)\left( \big(\mathbb{P}_h V_{h+1}^{\pi_b, \gamma}(s_h^i, a_h^i) - V_{h+1}^{\pi_b, \gamma}(s_{h+1}^i)\big) \right) \right\|_{(\Lambda_h + \lambda I)^{-1}}}_{\text{(vi)}}, \tag{23}$$

First, we bound (iv) and (v). By Theorem 3.3, we have

$$\mathbb{E}_{\mathcal{D}_h}\big[\|\widehat{\pi}_h(\cdot \mid s_h) - \pi_{b,h}(\cdot \mid s_h)\|_1^2\big] \leq \mathcal{O}\left( \frac{\log\big(H \cdot N(\mathcal{M}_h, \|\cdot\|_\infty, 1/n)/\delta\big)}{n} \right)$$

and

$$\mathbb{E}_{\mathcal{D}_h}\big[\|\widehat{Q}_h(s_h, \cdot) - Q_h^{\pi_b, \gamma}(s_h, \cdot)\|_1^2\big] \leq \mathcal{O}\left( \frac{H^2 e^{2H} \cdot |\mathcal{A}|^2 \cdot \log\big(H \cdot N(\mathcal{M}_h, \|\cdot\|_\infty, 1/n)/\delta\big)}{n} \right) \tag{24}$$

hold for every $h \in [H]$ with probability at least $1 - \delta/2$. By $\widehat{V}_h(s) = \langle \widehat{Q}_h(s, \cdot), \widehat{\pi}_h(\cdot \mid s)\rangle_{\mathcal{A}}$ for every $s_{h+1} \in \mathcal{S}$, we have

$$\mathbb{E}_{\mathcal{D}_h}\big[|\widehat{V}_{h+1}(s_{h+1}) - V_{h+1}^{\pi_b, \gamma}(s_{h+1})|^2\big] \leq \mathcal{O}\left( \frac{H^2 e^{2H} \cdot |\mathcal{A}|^2 \cdot \log\big(H \cdot N(\mathcal{M}_h, \|\cdot\|_\infty, 1/n)/\delta\big)}{n} \right) \tag{25}$$

for all $h \in [H]$ simultaneously. In the following proof, we will condition on these events. For notation simplicity, we define two function $f : \mathcal{X} \to \mathbb{R}$ and $\widehat{f} : \mathcal{X} \to \mathbb{R}$ for each $h \in [H]$, and dataset $\{x_i\}_{i \in [n]}$. We consider two cases: (1) $\widehat{f}_h = \widehat{Q}_h$, $f_h = Q_h^{\pi_b, \gamma}$, and $x_i = (s_h^i, a_h^i)$, $\mathcal{X} = \mathcal{S} \times \mathcal{A}$, (2) $\widehat{f}_h = \widehat{V}_{h+1}$, $f_h = V_{h+1}^{\pi_b, \gamma}$, and $x_i = s_{h+1}^i$, $\mathcal{X} = \mathcal{S}$. To bound (iv) and (v), we only need to uniformly bound

$$\left\| \sum_{i=1}^{n} \phi(s_h^i, a_h^i) \big(f_h(x_i) - \hat{f}_h(x_i)\big) \right\|_{(\Lambda_h + \lambda I)^{-1}}. \tag{26}$$

in both cases. We denote term $f_h(x_i) - \hat{f}_h(x_i)$ by $\epsilon_i$. Since we condition on equation 24 and equation 25, we have

$$\sum_{i=1}^{n} \epsilon_i^2 \le \mathcal{O}\left( H^2 e^{2H} \cdot |\mathcal{A}|^2 \cdot \log\big(H \cdot N(\mathcal{M}_h, \|\cdot\|_\infty, 1/n)/\delta\big) \right)$$

for both cases (1) and (2). Meanwhile, we also have

$$equation\ 26^2 = \left( \sum_{i=1}^{n} \epsilon_i \phi(s_h^i, a_h^i) \right)^\top \left( \lambda I + \sum_{i=1}^{n} \phi(s_h^i, a_h^i)\phi(s_h^i, a_h^i)^\top \right)^{-1} \left( \sum_{i=1}^{n} \epsilon_i \phi(s_h^i, a_h^i) \right)$$

$$= \mathrm{Tr}\left( \left( \sum_{i=1}^{n} \epsilon_i \phi(s_h^i, a_h^i) \right)\left( \sum_{i=1}^{n} \epsilon_i \phi(s_h^i, a_h^i) \right)^\top \left( \lambda I + \sum_{i=1}^{n} \phi(s_h^i, a_h^i)\phi(s_h^i, a_h^i)^\top \right)^{-1} \right).$$

By Lemma F.2, we have

$$\left( \sum_{i=1}^{n} \phi(s_h^i, a_h^i)\epsilon_i \right)\left( \sum_{i=1}^{n} \phi(s_h^i, a_h^i)\epsilon_i \right)^\top \le \mathcal{O}\left( H^2 e^{2H} \cdot |\mathcal{A}|^2 \cdot \log(H \cdot N(\Theta, \|\cdot\|_\infty, 1/n)/\delta) \right) \cdot \left( \sum_{i=1}^{n} \phi(s_h^i, a_h^i)\phi(s_h^i, a_h^i)^\top \right),$$

and therefore we have

$$\mathrm{Tr}\left( \left( \sum_{i=1}^{n} \epsilon_i \phi(s_h^i, a_h^i) \right)\left( \sum_{i=1}^{n} \epsilon_i \phi(s_h^i, a_h^i) \right)^\top \left( \lambda I + \sum_{i=1}^{n} \phi(s_h^i, a_h^i)\phi(s_h^i, a_h^i)^\top \right)^{-1} \right)$$

$$\le \mathcal{O}\left( H^2 e^{2H} \cdot |\mathcal{A}| \cdot \log(N(\Theta, \|\cdot\|_\infty, 1/n)/\delta) \right)$$

$$\cdot \mathrm{Tr}\left( \left( \sum_{i=1}^{n} \phi(s_h^i, a_h^i)\phi(s_h^i, a_h^i)^\top + \lambda \right)^{-1} \left( \sum_{i=1}^{n} \phi(s_h^i, a_h^i)\phi(s_h^i, a_h^i)^\top \right) \right)$$

$$\le d \cdot H^2 e^{2H} \cdot |\mathcal{A}|^2 \log(N(\Theta, \|\cdot\|_\infty, 1/n)/\delta). \tag{27}$$

here the last inequality comes from Lemma F.1. Therefore we have

$$(\mathrm{iii}) \le \sqrt{d} H e^H \cdot |\mathcal{A}| \cdot \sqrt{\log(N(\Theta, \|\cdot\|_\infty, 1/n)/\delta)}$$

Next, we bound (vi). We prove the following Lemma:

**Lemma C.1.** *Let $V : \mathcal{S} \to [0, H]$ be any fixed function. With our dataset $\mathcal{D} = \{\mathcal{D}_h\}_{h \in [H]}$, we have*

$$\left\| \sum_{i=1}^{n} \phi(s_h^i, a_h^i)\Big( \big(\mathbb{P}_h V_{h+1}(s_h^i, a_h^i) - V_{h+1}(s_{h+1}^i)\big) \Big) \right\|_{(\Lambda_h + \lambda I)^{-1}}^2 \le H^2 \cdot \big(2 \cdot \log(H/\delta) + d \cdot \log(1 + n/\lambda)\big)$$

*with probability at least $1 - \delta$ for all $h \in [H]$.*

*Proof.* Note that for $V_{h+1} : \mathcal{S} \to \mathbb{R}$, we have

$$\mathbb{E}[V_{h+1}(s_{h+1}^i) - \mathbb{P}_h V_{h+1}(s_h^i, a_h^i) \mid \mathcal{F}_h^i] = \mathbb{E}[V_{h+1}(s_{h+1}^i) - \mathbb{P}_h V_{h+1}(s_h^i, a_h^i) \mid s_h^i, a_h^i] = 0.$$

Here $\mathcal{F}_h^i = \sigma\big(\{(s_t^i, a_t^i\}_{t=1}^h)$ is the filtration generated by state-action pair before step $h+1$ for the $i$-th trajectory. We now invoke Lemma F.4 with $M_0 = \lambda I$ and $M_n = \lambda I + \sum_{i=1}^n \phi(s_h^i, a_h^i)\phi(s_h^i, a_h^i)^\top$. We have

$$\left\| \sum_{i=1}^{n} \phi(s_h^i, a_h^i)\Big( \big(\mathbb{P}_h V_{h+1}(s_h^i, a_h^i) - V_{h+1}(s_{h+1}^i)\big) \Big) \right\|_{(\Lambda_h + \lambda I)^{-1}}^2$$

$$\le 2H^2 \cdot \log\left( H \cdot \frac{\det(\Lambda_h + \lambda I)^{1/2}}{\delta \cdot \det(\lambda I)^{1/2}} \right)$$

with probability at least $1 - \delta/2$. Recall that by Assumption 3.4, $\|\phi(s, a)\|_2 \leq 1$ and therefore we have $\det(\Lambda_h + \lambda I) \leq (\lambda + n)^d$. Also we have $\det(\lambda I) = \lambda^d$, and we have

$$
\left\| \sum_{i=1}^{n} \phi(s_h^i, a_h^i) \Big( \big( \mathbb{P}_h V_{h+1}(s_h^i, a_h^i) - V_{h+1}(s_{h+1}^i) \big) \Big) \right\|_{(\Lambda_h + \lambda I)^{-1}}^2
$$
$$
\leq H^2 \cdot \big( 2 \cdot \log(H/\delta) + d \cdot \log(1 + n/\lambda) \big).
$$

$\square$

By equation 34, equation 38, Lemma C.1 and Assumption 3.4, we have

$$
\text{(iii)} \leq \mathcal{O}\bigg( (1+\gamma) \cdot \sqrt{d \cdot H^2 e^{2H} \cdot |\mathcal{A}| \cdot \log \big( H \cdot N(\Theta, \| \cdot \|_\infty, 1/n)/\delta \big)} \bigg) \leq \mathcal{O}\bigg( (1+\gamma) \cdot d H e^H \sqrt{|\mathcal{A}| \log(nH/\lambda\delta)} \bigg).
$$

Therefore (ii) $\leq \mathcal{O}\bigg( (1 + \gamma) \cdot |\mathcal{A}| \cdot d \cdot H e^H \sqrt{\log(nH/\lambda\delta)} \bigg) \cdot \|\phi(s, a)\|_{(\Lambda_h + \lambda I)^{-1}}$. Combined with equation 22, we conclude the proof of Theorem 3.5.

## D  PROOF FOR THEOREM 4.3

In this section, we prove Theorem 4.3. First, we invoke the following theorem, whose proof can be found in Jin et al. (2021), Appendix 5.2.

**Theorem D.1** (Theorem 4.2 in Jin et al. (2021))**.** *Suppose $\{\Gamma_h\}_{h=1}^{H}$ in Algorithm 2 is a uncertainty quantifier defined in equation 7. Under the event which equation 7 holds, suboptimality of Algorithm 2 satisfies*

$$
\text{SubOpt}(\{\tilde{\pi}_h\}_{h \in [H]}) \leq 2 \sum_{h=1}^{H} \mathbb{E}_{\pi^*} \left[ \Gamma_h(s_h, a_h) \right].
$$

*Here $\mathbb{E}_{\pi^*}$ is with respect to the trajectory induced by $\pi^*$ in the underlying MDP.*

With Theorem equation D.1, our proof for Theorem equation 4.3 then proceeds in **two steps**: (1) We prove that our uncertainty quantifier defined in 8, with $\beta$ defined in 2, is an uncertainty quantifier, with probability at least $1 - \delta/2$; (2) We prove that with penalty function set in 8, we can bound $\sum_{h=1}^{H} \mathbb{E}_{\pi^*} \left[ \Gamma_h(s_h, a_h) \right]$ with probability at least $1 - \delta/2$.

**Step (1).**  We now prove that $\Gamma_h$ defined in 8 is an uncertainty quantifier, with

$$
\beta = \mathcal{O}(H e^H \cdot |\mathcal{A}| \cdot d \sqrt{\log \big( nH/\delta \big)}).
$$

We have

$$
\Big| \big( \widehat{r}_h + \widetilde{\mathbb{P}}_h \widetilde{V}_{h+1} \big)(s, a) - \big( r_h + \mathbb{P}_h \widetilde{V}_{h+1} \big)(s, a) \Big|
$$
$$
\leq \underbrace{\big| \widehat{r}_h(s, a) - r_h(s, a) \big|}_{\text{(i)}} + \underbrace{\big| \widetilde{\mathbb{P}}_h \widetilde{V}_{h+1}(s, a) - \mathbb{P}_h \widetilde{V}_{h+1}(s, a) \big|}_{\text{(ii)}},
$$

To bound (i), recall that we construct $\widehat{r}_h$ by Algorithm 1 with guarantee

$$
|r_h(s, a) - \widehat{r}_h(s, a)| \leq \|\phi(s, a)\|_{(\Lambda_h + \lambda I)^{-1}} \cdot \mathcal{O}\big( (1 + \gamma) \cdot H e^H |\mathcal{A}| \cdot d \sqrt{\log(nH/\delta)} \big)
$$

for all $(s, a) \in \mathcal{S} \times \mathcal{A}$ with $\lambda = 1$. To bound (ii), recall that we construct $\widetilde{\mathbb{P}}_h \widetilde{V}_{h+1}(s, a) = \phi(s, a) \cdot \widetilde{u}_h$ by the Algorithm 2,

$$
\widetilde{u}_h = \text{argmin}_u \sum_{i=1}^{n} \big( \phi(s_h^i, a_h^i) \cdot u - \widetilde{V}_{h+1}(s_{h+1}^i) \big)^2 + \lambda \cdot \|u\|^2,
$$

note that we have a closed form solution for $\widetilde{u}_h$,

$$
\widetilde{u}_h = \big( \Lambda_h + \lambda I \big)^{-1} \bigg( \sum_{i=1}^{n} \phi(s_h^i, a_h^i) \widetilde{V}_{h+1}(s_{h+1}^i) \bigg),
$$

And by Assumption 4.1, we have $\mathbb{P}_h \widetilde{V}_{h+1}(s,a) = \phi(s,a) \cdot u_h$ with $\|u_h\| \leq (H - h + 1)\sqrt{d}$, therefore we have

$$
\begin{aligned}
\left| \widetilde{\mathbb{P}}_h \widetilde{V}_{h+1}(s,a) - \mathbb{P}_h \widetilde{V}_{h+1}(s,a) \right| &= \left| \phi(s,a)(u_h - \widetilde{u}_h) \right| \\
&= \left| \phi(s,a)(\Lambda_h + \lambda I)^{-1} \left( \sum_{i=1}^{n} \phi(s_h^i, a_h^i)\big(\widetilde{V}_{h+1}(s_{h+1}^i) - \mathbb{P}_h \widetilde{V}_{h+1}(s_h^i, a_h^i)\big) \right) \right. \\
&\qquad \left. + \phi(s,a)(\Lambda_h + \lambda I)^{-1} u_h \right| \\
&\leq \left| \phi(s,a)(\Lambda_h + \lambda I)^{-1} \left( \sum_{i=1}^{n} \phi(s_h^i, a_h^i)\big(\widetilde{V}_{h+1}(s_{h+1}^i) - \mathbb{P}_h \widetilde{V}_{h+1}(s_h^i, a_h^i)\big) \right) \right| \\
&\qquad + \lambda \cdot \left| \phi(s,a)(\Lambda_h + \lambda I)^{-1} u_h \right|,
\end{aligned}
$$

and with Caucht-Schwarz inequality we have

$$
\begin{aligned}
\left| \widetilde{\mathbb{P}}_h \widetilde{V}_{h+1}(s,a) - \mathbb{P}_h \widetilde{V}_{h+1}(s,a) \right| &\leq \|\phi(s,a)\|_{(\Lambda_h + \lambda I)^{-1}} \cdot \left( \lambda \|u_h\|_{(\Lambda_h + \lambda I)^{-1}} \right. \\
&\qquad \left. + \left\| \left( \sum_{i=1}^{n} \phi(s_h^i, a_h^i)\big(\widetilde{V}_{h+1}(s_{h+1}^i) - \mathbb{P}_h \widetilde{V}_{h+1}(s_h^i, a_h^i)\big) \right) \right\|_{(\Lambda_h + \lambda I)^{-1}} \right) \\
&\leq \|\phi(s,a)\|_{(\Lambda_h + \lambda I)^{-1}} \cdot \left( H\sqrt{\lambda d} \right. \\
&\qquad \left. + \underbrace{\left\| \left( \sum_{i=1}^{n} \phi(s_h^i, a_h^i)\big(\widetilde{V}_{h+1}(s_{h+1}^i) - \mathbb{P}_h \widetilde{V}_{h+1}(s_h^i, a_h^i)\big) \right) \right\|_{(\Lambda_h + \lambda I)^{-1}}}_{\text{(iii)}} \right).
\end{aligned}
$$

Completing the first step now suffices to bound (iii). However, (iii) is a self-normalizing summation term with $\widetilde{V}_{h+1}$ depends on dataset $\{(s_t^i, a_t^i)\}_{t > h, i \in [n]}$, therefore we cannot directly use Lemma F.4. We first prove the following lemma, which bound $\|\widetilde{u}_h + \widehat{w}_h\|$.

**Lemma D.2.** *In Algorithm 2, we have*

$$
\|\widetilde{u}_h + \widehat{w}_h\| \leq 2H\sqrt{nd/\lambda}.
$$

*Proof.* For the proof we only need to bound $\|\widetilde{u}_h\|$ and $\|\widehat{w}_h\|$ respectively. First we have

$$
\begin{aligned}
\|\widetilde{u}_h\| &= \left\| (\Lambda_h + \lambda I)^{-1} \left( \sum_{i=1}^{n} \phi(s_h^i, a_h^i) \widetilde{V}_{h+1}(s_{h+1}^i) \right) \right\| \\
&\leq H \cdot \sum_{i=1}^{n} \left\| (\Lambda_h + \lambda I)^{-1} \phi(s_h^i, a_h^i) \right\| \\
&\leq H\sqrt{n/\lambda} \cdot \sqrt{\operatorname{Tr}\left( (\Lambda_h + \lambda I)^{-1} \Lambda_h \right)} \\
&\leq H\sqrt{nd/\lambda}.
\end{aligned}
$$

Here the first inequality comes from $\widetilde{V}_{h+1} \in [0, H]$, and the second inequality comes from Jensen's inequality. Similarly, we have

$$
\begin{aligned}
\|\widehat{w}_h\| &= \left\| (\Lambda_h + \lambda I)^{-1} \left( \sum_{i=1}^{n} \phi(s_h^i, a_h^i)\big(\widehat{Q}_h(s_h^i, a_h^i) - \widehat{V}_{h+1}(s_{h+1}^i)\big) \right) \right\| \\
&\leq H\sqrt{nd/\lambda}.
\end{aligned}
$$

Therefore we complete the proof. $\qquad\square$

With $\|\widetilde{u}_h + \widehat{w}_h\|$ bounded, we can now invoke Theorem F.6 to bound term (iii). Set $R_0 = 2H\sqrt{nd/\lambda}$, $B = 2\beta$, $\lambda = 1$ and $\epsilon = dH/n$, we have

$$
\begin{aligned}
\text{(iii)} &\leq \sup_{V \in \mathcal{V}_{h+1}(R,B,\lambda)} \left\| \sum_{i=1}^{n} \phi\left(s_h^i, a_h^i\right) \cdot \left(V(s_{h+1}^i) - \mathbb{E}[V(s_{h+1}) \mid s_h^i, a_h^i])\right) \right\|_{(\Lambda_h + \lambda I)^{-1}} \\
&\leq \mathcal{O}(dH \cdot \log(dHn/\delta)).
\end{aligned}
$$

Therefore we have

$$
\left|\widetilde{\mathbb{P}}_h \widetilde{V}_{h+1}(s,a) - \mathbb{P}_h \widetilde{V}_{h+1}(s,a)\right| \leq \|\phi(s,a)\|_{(\Lambda_h + \lambda I)^{-1}} \cdot \mathcal{O}\left(dH \cdot \log(dHn/\delta) + H\sqrt{d}\right).
$$

Set $\lambda = 1$ in Theorem 3.5, we have

$$
|r_h(s,a) - \widehat{r}_h(s,a)| + \left|\widetilde{\mathbb{P}}_h \widetilde{V}_{h+1}(s,a) - \mathbb{P}_h \widetilde{V}_{h+1}(s,a)\right| \leq \beta \|\phi(s,a)\|_{(\Lambda_h + \lambda I)^{-1}} \tag{28}
$$

holds with probability at least $1 - \delta$. Recall that $\Gamma_h = \beta \|\phi(s,a)\|_{(\Lambda_h + \lambda I)^{-1}}$, we prove that $\Gamma_h$ is an uncertainty quantifier defined in 7. To finish the proof of Theorem 4.3, it suffices to finish the proof of the second step, i.e., we bound the term

$$
\sum_{i=1}^{n} \mathbb{E}_{\pi^*}[\Gamma_h(s_h, a_h)] = \sum_{i=1}^{n} \beta \cdot \mathbb{E}_{\pi^*}[\|\phi(s_h, a_h)\|_{(\Lambda_h + \lambda I)^{-1}}].
$$

**Step (2).** By Cauchy-Schwarz inequality, we have

$$
\begin{aligned}
&\mathbb{E}_{\pi^*}\left[\left(\phi\left(s_h, a_h\right)^\top \left(\Lambda_h + \lambda I\right)^{-1}\phi\left(s_h, a_h\right)\right)^{1/2}\right] \\
&= \mathbb{E}_{\pi^*}\left[\sqrt{\text{Tr}\left(\phi\left(s_h, a_h\right)^\top \left(\Lambda_h + \lambda I\right)^{-1}\phi\left(s_h, a_h\right)\right)}\right] \\
&= \mathbb{E}_{\pi^*}\left[\sqrt{\text{Tr}\left(\phi\left(s_h, a_h\right)\phi\left(s_h, a_h\right)^\top \left(\Lambda_h + \lambda I\right)^{-1}\right)}\right] \\
&\leq \sqrt{\text{Tr}\left(\mathbb{E}_{\pi^*}\left[\phi\left(s_h, a_h\right)\phi\left(s_h, a_h\right)^\top\right]\Lambda_h^{-1}\right)}
\end{aligned} \tag{29}
$$

for all $h \in [H]$. For notational simplicity, we define

$$
\Sigma_h = \mathbb{E}_{\pi^*}\left[\phi\left(s_h, a_h\right)\phi\left(s_h, a_h\right)^\top\right]
$$

for all $h \in [H]$. Condition on the event in Equation equation 28 and with Assumption 4.2, we have

$$
\begin{aligned}
\text{SubOpt}\left(\{\widetilde{\pi}_h\}_{h \in [H]}\right) &\leq 2\beta \cdot \sum_{h=1}^{H} \mathbb{E}_{\pi^*}\left[\phi(s_h, a_h)^\top (\Lambda_h + \lambda \cdot I)^{-1}\phi(s_h, a_h)\right] \\
&\leq 2\beta \sum_{h=1}^{H} \sqrt{\text{Tr}\left(\Sigma_h \cdot (I + c^\dagger \cdot n \cdot \Sigma_h)^{-1}\right)} \\
&= 2\beta \sum_{h=1}^{H} \sqrt{\sum_{j=1}^{d} \frac{\lambda_{h,j}}{1 + c^\dagger \cdot n \cdot \lambda_{h,j}}},
\end{aligned}
$$

here $\{\lambda_{h,j}\}_{j=1}^{d}$ are the eigenvalues of $\Sigma_h$. The first inequality comes from the event in Equation equation 28, the second inequality comes from Equation equation 29. Meanwhile, by Assumption 3.4, we have $\|\phi(s,a)\| \leq 1$ for all $(s,a) \in \mathcal{S} \times \mathcal{A}$. By Jensen's inequality, we have

$$
\|\Sigma_h\|_2 \leq \mathbb{E}_{\pi^*}\left[\|\phi\left(s_h, a_h\right)\phi\left(s_h, a_h\right)^\top\|_2\right] \leq 1
$$

for all $h \in [H]$, for all $s_h \in \mathcal{S}$ and all $h \in [H]$. As $\Sigma_h$ is positive semidefinite, we have $\lambda_{h,j} \in [0, 1]$ for all $x \in \mathcal{S}$, all $h \in [H]$, and all $j \in [d]$. Hence, we have

$$
\begin{aligned}
\text{SubOpt}(\{\widetilde{\pi}_h\}_{h \in [H]}) &\leq 2\beta \sum_{h=1}^{H} \sqrt{\sum_{j=1}^{d} \frac{\lambda_{h,j}}{1 + c^\dagger \cdot n \cdot \lambda_{h,j}}} \\
&\leq 2\beta \sum_{h=1}^{H} \sqrt{\sum_{j=1}^{d} \frac{1}{1 + c^\dagger \cdot n}} \leq c' \cdot d^{3/2}H^2n^{-1/2}\sqrt{\xi},
\end{aligned}
$$

where $\xi = \sqrt{\log(dHn/\delta)}$, the second inequality follows from the fact that $\lambda_{h,j} \in [0,1]$ for all $h \in [H]$, and all $j \in [d]$, while the third inequality follows from the choice of the scaling parameter $\beta > 0$ in Theorem 4.3. Here we define the absolute constant $c' = 2c/\sqrt{c^\dagger} > 0$, where $c^\dagger > 0$ is the absolute constant used in Assumption 4.2. Moreover, we consider the case of $\text{rank}(\Sigma_h) \leq r$. Then we have

$$
\begin{aligned}
\text{SubOpt}(\{\widetilde{\pi}_h\}_{h\in[H]}) &\leq 2\beta \sum_{h=1}^{H} \sqrt{\sum_{j=1}^{d} \frac{\lambda_{h,j}}{1 + c^\dagger \cdot n \cdot \lambda_{h,j}}} \\
&= 2\beta \sum_{h=1}^{H} \sqrt{\sum_{j=1}^{r} \frac{\lambda_{h,j}}{1 + c^\dagger \cdot n \cdot \lambda_{h,j}}} \\
&\leq 2\beta \sum_{h=1}^{H} \sqrt{\sum_{j=1}^{r} \frac{1}{1 + c^\dagger \cdot n}} \leq c' \cdot r^{1/2} dH^2 n^{-1/2} \sqrt{\xi}
\end{aligned}
$$

Thus we finish the proof for Theorem 4.3.

## E   PROOF FOR RKHS CASE

In this section, we prove the results of DCPPO in RKHS model class. In the following, we adopt an equivalent set of notations for ease of presentation. We formally write the inner product in $\mathcal{H}$ as $\langle f, f' \rangle_{\mathcal{H}} = f^\top f' = f'^\top f$ for any $f, f' \in \mathcal{H}$, so that $f(z) = \langle \phi(z), f \rangle_{\mathcal{H}} = f^\top \phi(z)$ for any $f \in \mathcal{H}$ and any $z \in \mathcal{Z}$. Moreover we denote the operators $\Phi_h : \mathcal{H} \to \mathbb{R}^n$ and $\Lambda_h : \mathcal{H} \to \mathcal{H}$ as

$$
\Phi_h = \begin{pmatrix} \phi\left(z_h^1\right)^\top \\ \vdots \\ \phi\left(z_h^n\right)^\top \end{pmatrix}, \quad \Lambda_h = \lambda \cdot I_{\mathcal{H}} + \sum_{i\in[n]} \phi\left(z_h^\tau\right) \phi\left(z_h^\tau\right)^\top = \lambda \cdot I_{\mathcal{H}} + \Phi_h^\top \Phi_h
$$

where $I_{\mathcal{H}}$ is the identity mapping in $\mathcal{H}$ and all the formal matrix multiplications follow the same rules as those for real-valued matrix. In this way, these operators are well-defined. Also, $\Lambda_h$ is a self-adjoint operator eigenvalue no smaller than $\lambda$, in the sense that $\langle f, \Lambda_h g \rangle = \langle \Lambda_h f, g \rangle$ for any $f, g \in \mathcal{H}$. Therefore, there exists a positive definite operator $\Lambda_h^{1/2}$ whose eigenvalues are no smaller than $\lambda^{1/2}$ and $\Lambda_h = \Lambda_h^{1/2} \Lambda_h^{1/2}$. We denote the inverse of $\Lambda_h^{1/2}$ as $\Lambda_h^{-1/2}$, so that $\Lambda_h^{-1} = \Lambda_h^{-1/2} \Lambda_h^{-1/2}$ and $\left\| \Lambda_h^{-1/2} \right\|_{\mathcal{H}} \leq \lambda^{-1/2}$. For any $z \in \mathcal{Z}$, we denote $\Lambda_h(z) = \Lambda_h + \phi(z)\phi(z)^\top$. In particular, it holds that

$$
\left( \Phi_h^\top \Phi_h + \lambda \cdot I_{\mathcal{H}} \right) \Phi_h^\top = \Phi_h^\top \left( \Phi_h \Phi_h^\top + \lambda \cdot I \right).
$$

Since both the matrix $\Phi_h \Phi_h^\top + \lambda \cdot I$ and the operator $\Phi_h^\top \Phi_h + \lambda \cdot I_{\mathcal{H}}$ are strictly positive definite, we have

$$
\Phi_h^\top \left( \Phi_h \Phi_h^\top + \lambda \cdot I \right)^{-1} = \left( \Phi_h^\top \Phi_h + \lambda \cdot I_{\mathcal{H}} \right)^{-1} \Phi_h^\top. \tag{30}
$$

Our learning process would depend on the "complexity" of the dataset sampled by $\pi_b$. To measure this complexity, we make the following definition.

**Definition E.1** (Effective dimension). *For all $h \in [H]$, Denote $\Sigma_h = \mathbb{E}_{\pi^b}\left[\phi\left(z_h\right)\phi\left(z_h\right)^\top\right], \Sigma_h^* = \mathbb{E}_{\pi^*}\left[\phi\left(z_h\right)\phi\left(z_h\right)^\top\right]$, where $\mathbb{E}_{\pi^*}$ is taken with respect to $(s_h, a_h)$ induced by the optimal policy $\pi^*$, and $\mathbb{E}_{\pi^b}$ is similarly induced by the behavior policy $\pi^b$. We define the (sample) effective dimension as*

$$
d_{eff}^{sample} = \sum_{h=1}^{H} \text{Tr}\left( \left(\Lambda_h + \lambda \mathcal{I}_{\mathcal{H}}\right)^{-1} \Sigma_h \right)^{1/2}.
$$

*Moreover, we define the population effective dimension under $\pi^b$ as*

$$
d_{eff}^{pop} = \sum_{h=1}^{H} \text{Tr}\left( \left(n \cdot \Sigma_h + \lambda \mathcal{I}_{\mathcal{H}}\right)^{-1} \Sigma_h^* \right)^{1/2}.
$$

We first present our result for reward estimation in the RKHS case:

**Theorem E.2** (**Reward Estimation for RKHS**). *For Algorithm 1 and 2, with probability at least $1 - \delta$, we have the following estimations of our reward function for all $z \in \mathcal{Z} \times \mathcal{A}$ and $\lambda > 1$,*

$$|r_h(z) - \widehat{r}_h(z)| \leq \|\phi(z)\|_{(\Lambda_h + \lambda \mathcal{I}_{\mathcal{H}})^{-1}} \cdot \mathcal{O}\left( H^2 \cdot G\left(n, 1 + 1/n\right) + \lambda \cdot R_r^2 + \zeta^2 \right)^{1/2},$$

*where $\zeta = \mathcal{O}\left( d_{eff}^{sample} \sqrt{\log\left( H \cdot N(\mathcal{Q}, \|\cdot\|_\infty, 1/n)/\delta \right)} \cdot H e^H \right)$, here $d_{eff}^{sample}$ is the sampling effective dimension.*

*Proof.* See Appendix E.1 for detailed proof. $\square$

Here we use the notation $\|\phi(z)\|_{(\Lambda_h + \lambda \mathcal{I}_{\mathcal{H}})^{-1}} = \langle \phi(z), (\Lambda_h + \lambda \mathcal{I}_{\mathcal{H}})^{-1}\phi(z)\rangle_{\mathcal{H}}$, where we define $\Lambda_h = \sum_{i=1}^n \phi(z_h^i)\phi(z_h^i)^\top$.

### E.1 PROOF FOR THEOREM E.2

Our proof for reward estimation in RKHS model class is very similar to the proof of linear model MDP, which can be found in Section C. To prove Theorem E.2, we first invoke Theorem 3.3, and we have

$$\mathbb{E}_{\mathcal{D}_h}\left[ \|\widehat{\pi}_h(\cdot \mid s_h) - \pi_{b,h}(\cdot \mid s_h)\|_1^2 \right] \leq \mathcal{O}\left( \frac{\log\left( H \cdot N(\mathcal{Q}, \|\cdot\|_\infty, 1/n)/\delta \right)}{n} \right)$$

and

$$\mathbb{E}_{\mathcal{D}_h}\left[ \|\widehat{Q}_h(s_h, \cdot) - Q_h^{\pi_b, \gamma}(s_h, \cdot)\|_1^2 \right] \leq \mathcal{O}\left( \frac{H^2 e^{2H} \cdot |\mathcal{A}|^2 \cdot \log\left( H \cdot N(\mathcal{Q}, \|\cdot\|_\infty, 1/n)/\delta \right)}{n} \right) \quad (31)$$

hold for every $h \in [H]$ with probability at least $1 - \delta/2$. Here the model class $\mathcal{Q}$ is defined in Assumption 5.2. Conditioning on this event, we have

$$\mathbb{E}_{\mathcal{D}_h}\left[ |\widehat{V}_{h+1}(s_{h+1}) - V_{h+1}^{\pi_b, \gamma}(s_{h+1})|^2 \right] \leq \mathcal{O}\left( \frac{H^2 e^{2H} \cdot |\mathcal{A}|^2 \cdot \log\left( H \cdot N(\mathcal{Q}, \|\cdot\|_\infty, 1/n)/\delta \right)}{n} \right) \tag{32}$$

for all $h \in [H]$ and all $s_{h+1} \in \mathcal{S}$ simultaneously. By Algorithm 1, we have

$$\widehat{r}_h = (\Lambda_h + \lambda I)^{-1}\left( \sum_{i=1}^n \phi(z_h^i)\left(\widehat{Q}_h(z_h^i) - \gamma \cdot \widehat{V}_{h+1}(s_{h+1}^i)\right) \right),$$

Recall that we denote $(s, a) \in \mathcal{S} \times \mathcal{A}$ by $z \in \mathcal{Z}$. Since we have $r_h(z) = \phi(z) \cdot r_h$. By our construction for $\widehat{r}_h$ in Algorithm 1, we therefore have

$$|r_h(z) - \widehat{r}_h(z)| = |\phi(z)(r_h - \widehat{r}_h)|$$

$$= \left| \phi(z)\left(\Lambda_h + \lambda I\right)^{-1}\left( \lambda \cdot r_h + \sum_{i=1}^n \phi(z_h^i)\left(\widehat{Q}_h(z_h^i) - \gamma \cdot \widehat{V}_{h+1}(s_{h+1}^i) - r_h(z_h^i)\right) \right) \right|$$

$$\leq \underbrace{\lambda \cdot |\phi(z)\left(\Lambda_h + \lambda I\right)^{-1} r_h|}_{(i)}$$

$$+ \underbrace{\left| \phi(z)\left(\Lambda_h + \lambda I\right)^{-1}\left( \sum_{i=1}^n \left(\widehat{Q}_h(z_h^i) - \gamma \cdot \widehat{V}_h(s_{h+1}^i) - r_h(z_h^i)\right) \right) \right|}_{(ii)}$$

holds for all $z \in \mathcal{Z}$. For (i), we have

$$(i) \leq \lambda \cdot \|\phi(z)\|_{(\Lambda_h + \lambda I)^{-1}} \cdot \|r_h\|_{(\Lambda_h + \lambda I)^{-1}},$$

by Cauchy-Schwarz inequality and by $\Lambda_h$ being semi-positive definite and $\|r_h\|_{\mathcal{H}} \leq R_r$, we have

$$\text{(i)} \leq \lambda \cdot R_r/\sqrt{\lambda} \cdot \|\phi(z)\|_{(\Lambda_h+\lambda I)^{-1}} = \sqrt{\lambda} \cdot R_r \cdot \|\phi(z)\|_{(\Lambda_h+\lambda I)^{-1}}, \tag{33}$$

and

$$\text{(ii)} \leq \|\phi(z)\|_{(\Lambda_h+\lambda I)^{-1}} \cdot \underbrace{\left\| \sum_{i=1}^{n} \phi(z_h^i)\big(\widehat{Q}_h(z_h^i) - \gamma \cdot \widehat{V}_{h+1}(s_{h+1}^i) - r_h(z_h^i)\big) \right\|_{(\Lambda_h+\lambda I)^{-1}}}_{\text{(iii)}}.$$

Recall that in equation 2, we have the following Bellman equation hold for all $(s_h, a_h) \in \mathcal{S} \times \mathcal{A}$,

$$r_h(z_h) + \gamma \cdot \mathbb{P}_h V_{h+1}^{\pi_b,\gamma}(z_h) = Q_h^{\pi_b,\gamma}(z_h),$$

substitute this into (iii), and we have

$$\text{(iii)} = \left\| \sum_{i=1}^{n} \phi(z_h^i)\Big( \big(\widehat{Q}_h(z_h^i) - Q_h^{\pi_b,\gamma}(z_h^i)\big) - \gamma \cdot \big(\widehat{V}_{h+1}(s_{h+1}^i) - \mathbb{P}_h V_{h+1}^{\pi_b,\gamma}(z_h^i)\big) \Big) \right\|_{(\Lambda_h+\lambda I)^{-1}}$$

$$\leq \underbrace{\left\| \sum_{i=1}^{n} \phi(z_h^i)\Big( \big(\widehat{Q}_h(z_h^i) - Q_h^{\pi_b,\gamma}(z_h^i)\big) \Big) \right\|_{(\Lambda_h+\lambda I)^{-1}}}_{\text{(iv)}}$$

$$+ \gamma \cdot \underbrace{\left\| \sum_{i=1}^{n} \phi(z_h^i)\Big( \big(\widehat{V}_{h+1}(s_{h+1}^i) - V_{h+1}^{\pi_b,\gamma}(s_{h+1}^i)\big) \Big) \right\|_{(\Lambda_h+\lambda I)^{-1}}}_{\text{(v)}}$$

$$+ \gamma \cdot \underbrace{\left\| \sum_{i=1}^{n} \phi(z_h^i)\Big( \big(\mathbb{P}_h V_{h+1}^{\pi_b,\gamma}(z_h^i) - V_{h+1}^{\pi_b,\gamma}(s_{h+1}^i)\big) \Big) \right\|_{(\Lambda_h+\lambda I)^{-1}}}_{\text{(vi)}}, \tag{34}$$

First, we bound (iv) and (v). By Theorem 3.3, we have

$$\mathbb{E}_{\mathcal{D}_h}\big[\|\widehat{\pi}_h(\cdot \mid s_h) - \pi_{b,h}(\cdot \mid s_h)\|_1^2\big] \leq \mathcal{O}\left( \frac{\log\big(H \cdot N(\mathcal{Q}, \|\cdot\|_\infty, 1/n)/\delta\big)}{n} \right)$$

and

$$\mathbb{E}_{\mathcal{D}_h}\big[\|\widehat{Q}_h(s_h, \cdot) - Q_h^{\pi_b,\gamma}(s_h, \cdot)\|_1^2\big] \leq \mathcal{O}\left( \frac{H^2 e^{2H} \cdot |\mathcal{A}|^2 \cdot \log\big(H \cdot N(\mathcal{Q}, \|\cdot\|_\infty, 1/n)/\delta\big)}{n} \right) \tag{35}$$

hold for every $h \in [H]$ with probability at least $1 - \delta/2$. By $\widehat{V}_h(s) = \langle \widehat{Q}_h(s, \cdot), \widehat{\pi}_h(\cdot \mid s)\rangle_{\mathcal{A}}$ for every $s_{h+1} \in \mathcal{S}$, we have

$$\mathbb{E}_{\mathcal{D}_h}\big[|\widehat{V}_{h+1}(s_{h+1}) - V_{h+1}^{\pi_b,\gamma}(s_{h+1})|^2\big] \leq \mathcal{O}\left( \frac{H^2 e^{2H} \cdot |\mathcal{A}|^2 \cdot \log\big(H \cdot N(\mathcal{Q}, \|\cdot\|_\infty, 1/n)/\delta\big)}{n} \right) \tag{36}$$

for all $h \in [H]$ simultaneously. In the following proof, we will condition on these events. For notation simplicity, we define two function $f : \mathcal{X} \to \mathbb{R}$ and $\widehat{f} : \mathcal{X} \to \mathbb{R}$ for each $h \in [H]$, and dataset $\{x_i\}_{i\in[n]}$. We consider two cases: (1) $\widehat{f}_h = \widehat{Q}_h$, $f_h = Q_h^{\pi_b,\gamma}$, and $x_i = z_h^i$, $\mathcal{X} = \mathcal{Z}$, (2) $\widehat{f}_h = \widehat{V}_{h+1}$, $f_h = V_{h+1}^{\pi_b,\gamma}$, and $x_i = s_{h+1}^i$, $\mathcal{X} = \mathcal{S}$. To bound (iv) and (v), we only need to uniformly bound

$$\left\| \sum_{i=1}^{n} \phi(z_h^i)\big(f_h(x_i) - \hat{f}_h(x_i)\big) \right\|_{(\Lambda_h+\lambda I)^{-1}}. \tag{37}$$

in both cases. We denote term $f_h(x_i) - \hat{f}_h(x_i)$ by $\epsilon_i$. Recall that we condition on equation 24 and equation 25, we have

$$\sum_{i=1}^{n} \epsilon_i^2 \leq \mathcal{O}\left( \frac{H^2 e^{2H} \cdot |\mathcal{A}|^2 \cdot \log\big(H \cdot N(\mathcal{Q}, \|\cdot\|_\infty, 1/n)/\delta\big)}{n} \right)$$

for both cases (1) and (2). Meanwhile, we also have

$$
\begin{aligned}
\text{equation } 26^2 &= \left( \sum_{i=1}^{n} \epsilon_i \phi(z_h^i) \right)^{\top} \left( \lambda I + \sum_{i=1}^{n} \phi(z_h^i)\phi(z_h^i)^{\top} \right)^{-1} \left( \sum_{i=1}^{n} \epsilon_i \phi(z_h^i) \right) \\
&= \text{Tr}\left( \left( \sum_{i=1}^{n} \epsilon_i \phi(z_h^i) \right) \left( \sum_{i=1}^{n} \epsilon_i \phi(z_h^i) \right)^{\top} \left( \lambda I_{\mathcal{H}} + \Phi_h^{\top} \Phi_h \right)^{-1} \right).
\end{aligned}
$$

By Lemma F.2, we have

$$
\left( \sum_{i=1}^{n} \phi(z_h^i)\epsilon_i \right) \left( \sum_{i=1}^{n} \phi(z_h^i)\epsilon_i \right)^{\top} \leq \mathcal{O}\left( H^2 e^{2H} \cdot |\mathcal{A}|^2 \cdot \log(H \cdot N(\mathcal{Q}, \|\cdot\|_{\infty}, 1/n)/\delta) \right) \cdot \left( \sum_{i=1}^{n} \Phi_h^{\top} \Phi_h \right),
$$

For notation simplicity, denote $\phi(z_h)$ by $\boldsymbol{u}_i$,

$$
\begin{aligned}
\text{Tr}&\left( \left( \sum_{i=1}^{n} \epsilon_i \phi(z_h) \right) \left( \sum_{i=1}^{n} \epsilon_i \phi(z_h) \right)^{\top} \left( \lambda I_{\mathcal{H}} + \Phi_h^{\top} \Phi_h \right)^{-1} \right) \\
&\leq \mathcal{O}\Bigg( \left( H^2 e^{2H} \cdot |\mathcal{A}| \cdot \log(N(\mathcal{Q}, \|\cdot\|_{\infty}, 1/n)/\delta) \right) \\
&\qquad \cdot \text{Tr}\left( \left( \Phi_h^{\top} \Phi_h + \lambda \mathcal{I}_{\mathcal{H}} \right)^{-1} \left( \Phi_h^{\top} \Phi_h \right) \right) \\
&\leq d_{\text{eff}}^{\text{sample}^2} \cdot H^2 e^{2H} \cdot |\mathcal{A}|^2 \log(N(\mathcal{Q}, \|\cdot\|_{\infty}, 1/n)/\delta).
\end{aligned} \tag{38}
$$

here the last inequality comes from the definition of $d_{\text{eff}}^{\text{sample}}$ and Lemma D.3 in Jin et al. (2021). Since there is no distribution shift, the effective dimension can be bounded by a constant. Next, we bound (vi). We prove the following lemma, which is the RKHS version of Lemma C.1.

**Lemma E.3.** *Let $V : \mathcal{S} \to [0, H]$ be any fixed function. With our dataset $\mathcal{D} = \{\mathcal{D}_h\}_{h \in [H]}$, we have*

$$
\left\| \sum_{i=1}^{n} \phi(z_h^i) \left( \left( \mathbb{P}_h V_{h+1}(z_h^i) - V_{h+1}(s_{h+1}^i) \right) \right) \right\|_{(\Lambda_h + \lambda I)^{-1}}^2 \leq H^2 \cdot G(n, 1 + 1/n) + 2H^2 \cdot \log(H/\delta)
$$

*with probability at least $1 - \delta$ for all $h \in [H]$ when $1 + 1/n \leq \lambda$.*

*Proof.* Note that for $V_{h+1} : \mathcal{S} \to \mathbb{R}$, we have

$$
\mathbb{E}[V_{h+1}(s_{h+1}^i) - \mathbb{P}_h V_{h+1}(s_h^i, a_h^i) \mid \mathcal{F}_h^i] = \mathbb{E}[V_{h+1}(s_{h+1}^i) - \mathbb{P}_h V_{h+1}(s_h^i, a_h^i) \mid s_h^i, a_h^i] = 0.
$$

Here $\mathcal{F}_h^i = \sigma\left(\{(s_t^i, a_t^i)\}_{t=1}^h\right)$ is the filtration generated by state-action pair before step $h+1$. We now invoke Lemma F.5 with $\epsilon_h^i = V_{h+1}(s_{h+1}^i) - \mathbb{P}_h V_{h+1}(s_h^i, a_h^i)$ and $\sigma^2 = H^2$ since $\epsilon_h^i \in [-H, H]$ and it holds with probability at least $1 - \delta$ for all $h \in [H]$ that

$$
E_h^{\top} \left[ (K_h + \eta \cdot I)^{-1} + I \right]^{-1} E_h \tag{39}
$$

$$
\leq H^2 \cdot \log\det\left[ (1 + \eta) \cdot I + K_h \right] + 2H^2 \cdot \log(H/\delta) \tag{40}
$$

for any $\eta > 0$, where $E_h = (\epsilon_h^i)_{i \in [n]}^{\top}$. We now transform into the desired form,

$$
\begin{aligned}
\left\| \sum_{i=1}^{n} \phi(s_h^i, a_h^i) \left( \left( \mathbb{P}_h V_{h+1}(s_h^i, a_h^i) - V_{h+1}(s_{h+1}^i) \right) \right) \right\|_{(\Lambda_h + \lambda I)^{-1}}^2 \\
&= E_h^{\top} \Phi_h \left( \Phi_h^{\top} \Phi_h + \lambda \cdot I_{\mathcal{H}} \right)^{-1} \Phi_h^{\top} E_h \\
&= E_h^{\top} \Phi_h \Phi_h^{\top} \left( \Phi_h \Phi_h^{\top} + \lambda \cdot I \right)^{-1} E_h \\
&= E_h^{\top} K_h \left( K_h + \lambda \cdot I \right)^{-1} E_h \\
&= E_h^{\top} E_h - \lambda \cdot E_h^{\top} \left( K_h + \lambda \cdot I \right)^{-1} E_h \\
&= E_h^{\top} E_h - E_h^{\top} \left( K_h/\lambda + I \right)^{-1} E_h,
\end{aligned} \tag{41}
$$

where the first equality follows from the definition of $\Lambda_h$, the second equality from equation 30, and the third equality follows from the fact that $K_h = \Phi_h^\top \Phi_h$. Therefore for any $\underline{\lambda} > 1$ such that $\lambda \geq \underline{\lambda}$, it holds that

$$\left\| \sum_{i=1}^n \phi(s_h^i, a_h^i) \Big( \big( \mathbb{P}_h V_{h+1}(s_h^i, a_h^i) - V_{h+1}(s_{h+1}^i) \big) \Big) \right\|^2_{(\Lambda_h + \lambda I)^{-1}} \leq E_h^\top K_h (K_h + \underline{\lambda} \cdot I)^{-1} E_h.$$

For any $\eta > 0$, noting that $\big( (K_h + \eta \cdot I)^{-1} + I \big) (K_h + \eta \cdot I) = K_h + (1 + \eta) \cdot I$, we have

$$\big( (K_h + \eta \cdot I)^{-1} + I \big)^{-1} = (K_h + \eta \cdot I)(K_h + (1 + \eta) \cdot I)^{-1} \tag{42}$$

Meanwhile, taking $\eta = \underline{\lambda} - 1 > 0$, we have

$$E_h^\top K_h (K_h + \underline{\lambda} \cdot I)^{-1} E_h \leq E_h^\top (K_h + \eta \cdot I)(K_h + \underline{\lambda} \cdot I)^{-1} E_h$$
$$= E_h^\top \left[ (K_h + \eta \cdot I)^{-1} + I \right]^{-1} E_h,$$

where the second line follows from equation 42. For any fixed $\delta > 0$, now we know that

$$\left\| \sum_{i=1}^n \phi(s_h^i, a_h^i) \Big( \big( \mathbb{P}_h V_{h+1}(s_h^i, a_h^i) - V_{h+1}(s_{h+1}^i) \big) \Big) \right\|^2_{(\Lambda_h + \lambda I)^{-1}} \leq H^2 \cdot \log \det \left[ \underline{\lambda} \cdot I + K_h \right] + 2H^2 \cdot \log(H/\delta)$$
$$\leq H^2 \cdot G(n, \underline{\lambda}) + 2H^2 \cdot \log(H/\delta) \tag{43}$$

for all $h \in [H]$ with probability at least $1 - \delta$. $\qquad\square$

Combining equation 34, equation 38, Lemma E.3 and Assumption 3.4, and set $\underline{\lambda} = 1 + 1/n \leq \lambda$, we have

$$\text{(iii)} \leq \mathcal{O}\left( (1+\gamma) \cdot d_{\text{eff}}^{\text{sample}} \cdot H e^H \cdot |\mathcal{A}| \sqrt{\log(N(\mathcal{Q}, \|\cdot\|_\infty, 1/n)/\delta)} + \sqrt{H^2 \cdot G(n, 1 + 1/n) + 2H^2 \cdot \log(H/\delta)} \right),$$

Since (ii) $\leq$ (iii) $\cdot \|\phi(z)\|_{\Lambda_h + \lambda \mathcal{I}_\mathcal{H}}$, combined with the bound for (i) in equation 33, we conclude the proof of Theorem E.2.

### E.2 Proof for Theorem 5.3

To prove Theorem, we again invoke Theorem D.1. Our proof proceeds in two steps: (1) We prove that with $\beta$ set in Theorem 5.3, equation 13 is an uncertainty quantifier with high probability for every $h \in [H]$. (2) We prove that with penalty function set in equation 13, we can bound $\sum_{i=1}^n \mathbb{E}_{\pi^*}[\Gamma_h(z_h)]$.

**Step (1).** In this step we prove that with $\beta$ specified in Theorem 5.3, the penalty functions $\{\Gamma_h\}_{h \in [H]}$ are uncertainty quantifiers with high probability. By Algorithm 2, We have

$$\left| \big( \widehat{r}_h + \widetilde{\mathbb{P}}_h \widetilde{V}_{h+1} \big)(s, a) - \big( r_h + \mathbb{P}_h \widetilde{V}_{h+1} \big)(s, a) \right|$$
$$\leq \underbrace{\left| \widehat{r}_h(s, a) - r_h(s, a) \right|}_{\text{(i)}} + \underbrace{\left| \widetilde{\mathbb{P}}_h \widetilde{V}_{h+1}(s, a) - \mathbb{P}_h \widetilde{V}_{h+1}(s, a) \right|}_{\text{(ii)}},$$

To bound (i), recall that we construct $\widehat{r}_h$ by Algorithm 1 with guarantee

$$|r_h(s, a) - \widehat{r}_h(s, a)| \leq \|\phi(s, a)\|_{(\Lambda_h + \lambda I)^{-1}}$$
$$\cdot \mathcal{O}\left( (1 + \gamma) d_{\text{eff}}^{\text{sample}} H e^H |\mathcal{A}| \sqrt{\log(H \cdot N(\mathcal{Q}, \|\cdot\|_\infty, 1/n)/\delta) + G(n, 1 + 1/n)} + \lambda \cdot R_r^2 \right) \tag{44}$$

for all $(s, a) \in \mathcal{S} \times \mathcal{A}$, $h \in [H]$ with probability at least $1 - \delta/2$. To bound (ii), recall that we construct $\widetilde{\mathbb{P}}_h \widetilde{V}_{h+1}(s, a) = \phi(s, a) \cdot \widetilde{u}_h$ by the Algorithm 2,

$$\widetilde{f}_h = \text{argmin}_{f \in \mathcal{H}} \sum_{i=1}^n \big( \phi(z_h^i) \cdot f - \widetilde{V}_{h+1}(s_{h+1}^i) \big)^2 + \lambda \cdot \|f\|_\mathcal{H}^2,$$

note that by the Representer's theorem (see (Steinwart & Christmann, 2008)), we have a closed form solution for $\widetilde{f}_h$,

$$\widetilde{f}_h = \left(\Phi_h^\top \Phi_h + \lambda \mathcal{I}_{\mathcal{H}}\right)^{-1} \Phi_h^\top \widetilde{y}_h,$$

here we use the notation $\widetilde{y}_h = (\widetilde{V}_{h+1}(s_{h+1}^i))^\top$. Meanwhile, with Assumption 5.2, we have $\mathbb{P}_h \widetilde{V}_{h+1}(s,a) = \phi(s,a) \cdot f_h$ with $\|f_h\|_{\mathcal{H}} \le R_{\mathcal{Q}} H$, therefore we have

$$
\begin{aligned}
\left|\widetilde{\mathbb{P}}_h \widetilde{V}_{h+1}(s,a) - \mathbb{P}_h \widetilde{V}_{h+1}(s,a)\right| &= \left|\phi(s,a)\left(f_h - \widetilde{f}_h\right)\right| \\
&= \left|\phi(s,a)^\top \left(\Phi_h^\top \Phi_h + \lambda \cdot I_{\mathcal{H}}\right)^{-1} \Phi_h^\top \widetilde{y}_h - \phi(s,a)^\top f_h\right| \\
&= |\underbrace{\phi(s,a)^\top \left(\Phi_h^\top \Phi_h + \lambda \cdot I_{\mathcal{H}}\right)^{-1} \Phi_h^\top \Phi_h f_h - \phi(s,a)^\top f_h}_{(i)}| \\
&\quad + |\underbrace{\phi(s,a)^\top \left(\Phi_h^\top \Phi_h + \lambda \cdot I_{\mathcal{H}}\right)^{-1} \Phi_h^\top \left(\widetilde{y}_h - \Phi_h f_h\right)}_{(ii)}|.
\end{aligned}
$$

In the sequel, we bound terms (i) and (ii) separately. By the Cauchy-Schwarz inequality,

$$
\begin{aligned}
|(i)| &= \left|\phi(s,a)^\top \left(\Phi_h^\top \Phi_h + \lambda \cdot I_{\mathcal{H}}\right)^{-1} \Phi_h^\top \Phi_h f_h - \phi(s,a)^\top f_h\right| \\
&= \left|\phi(s,a)^\top \left(\Phi_h^\top \Phi_h + \lambda \cdot I_{\mathcal{H}}\right)^{-1} \left[\Phi_h^\top \Phi_h - \left(\Phi_h^\top \Phi_h + \lambda \cdot I_{\mathcal{H}}\right)\right] f_h\right| \\
&= \lambda \cdot \left|\phi(s,a)^\top (\Lambda_h + \lambda \mathcal{I}_{\mathcal{H}})^{-1} f_h\right| \\
&\le \lambda \cdot \left\|(\Lambda_h + \lambda \mathcal{I}_{\mathcal{H}})^{-1} \phi(x,a)\right\|_{\mathcal{H}} \cdot \|f_h\|_{\mathcal{H}},
\end{aligned}
$$

recall that we define $\Lambda_h = \Phi_h^\top \Phi_h$. Therefore, it holds that

$$
\begin{aligned}
|(i)| &\le \lambda^{1/2} \cdot \left\|\Lambda_h^{-1/2} \phi(s,a)\right\|_{\mathcal{H}} \cdot \|f_h\|_{\mathcal{H}} \\
&\le R_r H \cdot \lambda^{1/2} \cdot \|\phi(s,a)\|_{(\Lambda_h + \lambda \mathcal{I}_{\mathcal{H}})^{-1}}.
\end{aligned}
$$

Here the first inequality comes from $\Lambda_h + \lambda \mathcal{I}_{\mathcal{H}} \succeq \lambda \mathcal{I}_{\mathcal{H}}$, and the second inequality comes from Assumption 5.2, On the other hand, we have

$$
\begin{aligned}
(ii) &= \left|\phi(s,a)^\top \left(\Phi_h^\top \Phi_h + \lambda \cdot \mathcal{I}_{\mathcal{H}}\right)^{-1} \Phi_h^\top \left(\widetilde{y}_h - \Phi_h f_h\right)\right| \\
&= \left|\phi(s,a)^\top \left(\Phi_h^\top \Phi_h + \lambda \cdot \mathcal{I}_{\mathcal{H}}\right)^{-1} \left(\sum_{i=1}^n \phi(s_h^i, a_h^i)\left(\widetilde{V}_{h+1}(s_{h+1}^i) - \mathbb{P}_h \widetilde{V}_{h+1}(s_h^i, a_h^i)\right)\right)\right| \\
&\le \|\phi(s,a)\|_{(\Lambda_h + \lambda \mathcal{I}_{\mathcal{H}})^{-1}} \cdot \underbrace{\left\|\sum_{i=1}^n \phi(s_h^i, a_h^i)\left(\widetilde{V}_{h+1}(s_{h+1}^i) - \mathbb{P}_h \widetilde{V}_{h+1}(s_h^i, a_h^i)\right)\right\|_{(\Lambda_h + \lambda \mathcal{I}_{\mathcal{H}})^{-1}}}_{(iii)} \quad (45)
\end{aligned}
$$

where the last inequality comes from Cauchy-Schwarz inequality. In the sequel, we aim to bound (iii). We define $\mathcal{F}_h^i = \sigma\left(\{(s_t^i, a_t^i)\}_{t=1}^h\right)$ to be the filtration generated by state-action pair before step $h+1$. With Lemma E.3, we have

$$(iii)^2 \le \mathcal{O}\left(H^2 \cdot G(n, \underline{\lambda}) + 2H^2 \cdot \log(H/\delta)\right)$$

with probbaility at least $1 - \delta/2$. Therefore we have

$$\left|\widetilde{\mathbb{P}}_h \widetilde{V}_{h+1}(s,a) - \mathbb{P}_h \widetilde{V}_{h+1}(s,a)\right| \le \|\phi(s,a)\|_{(\Lambda_h + \lambda \mathcal{I}_{\mathcal{H}})^{-1}} \cdot \mathcal{O}\left(R_r^2 H^2 \cdot \lambda + H^2 \cdot G(n, \lambda) + 2H^2 \cdot \log(H/\delta)\right)^{1/2},$$

and combined with equation 44, we have

$$
\begin{aligned}
\left|\left(\widehat{r}_h(s,a) + \widetilde{\mathbb{P}}_h \widetilde{V}_h(s,a)\right) - \left(r_h(s,a) + \mathbb{P}_h \widetilde{V}_h(s,a)\right)\right| &\le \left|\widehat{r}_h(s,a) - r_h(s,a)\right| + \left|\widetilde{\mathbb{P}}_h \widetilde{V}_h(s,a) - \mathbb{P}_h \widetilde{V}_h(s,a)\right| \\
&\le \|\phi(s,a)\|_{(\Lambda_h + \lambda \mathcal{I}_{\mathcal{H}})^{-1}} \\
&\quad \cdot \mathcal{O}\left(\lambda R_{\mathcal{Q}}^2 H^2 + H^2 G(n, 1 + 1/n)\right. \\
&\qquad \left. + d_{\text{eff}}^{\text{sample}\,2} H^2 e^{2H} |\mathcal{A}|^2 \log(H \cdot N(\mathcal{Q}, \|\cdot\|_\infty, 1/n)/\delta)\right)^{1/2} \\
&\qquad\qquad\qquad\qquad\qquad\qquad\qquad\qquad\qquad\qquad\qquad (46)
\end{aligned}
$$

with probability at least $1 - \delta$ for all $h \in [H]$. For the constant term on the right-hand side of equation 46, we have the following guarantee:

**Lemma E.4.** *We have*

$$\lambda R_r^2 H^2 + H^2 G(n, 1 + 1/n) + d_{\text{eff}}^{sample^2} H^2 e^{2H} |\mathcal{A}|^2 \cdot \log(H \cdot N(\mathcal{Q}, \|\cdot\|_\infty, 1/n)/\delta) \leq \beta^2$$

*for the three eigenvalue decay conditions discussed in Assumption 5.1 and $\beta$ set in Theorem 5.3.*

*Proof.* See Appendix E.3 for details. $\square$

With Lemma E.4, we prove that $\beta \cdot \|\phi(s,a)\|_{(\Lambda_h + \lambda \mathcal{I}_\mathcal{H})^{-1}}$ is an uncertainty quantifier satisfying condition 7. Now we transform it into the desired form in equation 13. Note that

$$
\begin{aligned}
\|\phi(z)\|^2_{(\Lambda_h + \lambda \mathcal{I}_\mathcal{H})^{-1}} &= \phi(z)^\top (\Phi_h^\top \Phi_h + \lambda \mathcal{I}_\mathcal{H})^{-1} \phi(z) \\
&= \frac{1}{\lambda} \big[ \phi(z)^\top \phi(z) - \phi(z)^\top \Phi_h^\top \Phi_h (\Phi_h^\top \Phi_h + \lambda \mathcal{I}_\mathcal{H})^{-1} \phi(z) \big] \\
&= \frac{1}{\lambda} \big[ K(z,z) - \phi(z)^\top \Phi_h(z)^\top \cdot \Phi_h (\Phi_h^\top \Phi_h + \lambda \mathcal{I}_\mathcal{H})^{-1} \phi(z) \big] \\
&= \frac{1}{\lambda} [K(z,z) - k_h(z)^\top (K_h + \lambda I)^{-1} k_h(z)], \qquad (47)
\end{aligned}
$$

we conclude that

$$\Gamma_h(z) = \beta \cdot \lambda^{-1/2} \cdot (K(z,z) - k_h(z)^\top (K_h + \lambda I)^{-1} k_h(z))^{1/2}, \qquad (48)$$

and thus we complete the first step.

**Step (2).** The second step is to prove that with $\Gamma_h$ given by 13 and $\beta$ given by Theorem 5.3, we can give an upper bound for the suboptimality gap. Recall that for $z \in \mathcal{Z}$, we define $\Lambda_h(z) = \Lambda_h + \phi(z)\phi(z)^\top$, therefore we have

$$\Lambda_h(z) + \lambda \mathcal{I}_\mathcal{H} = (\Lambda_h + \lambda \mathcal{I}_\mathcal{H})^{1/2} \big( \mathcal{I}_\mathcal{H} + (\Lambda_h + \lambda \mathcal{I}_\mathcal{H})^{-1/2} \phi(z)\phi(z)^\top (\Lambda_h + \lambda \mathcal{I}_\mathcal{H})^{-1/2} \big)(\Lambda_h + \lambda \mathcal{I}_\mathcal{H})^{1/2},$$

which indicates

$$
\begin{aligned}
\log \det((\Lambda_h(z) + \lambda \mathcal{I}_\mathcal{H})) &= \log \det((\Lambda_h + \lambda \mathcal{I}_\mathcal{H})) + \log \det \big( \mathcal{I}_\mathcal{H} + (\Lambda_h + \lambda \mathcal{I}_\mathcal{H})^{-1/2} \phi(z)\phi(z)^\top (\Lambda_h + \lambda \mathcal{I}_\mathcal{H})^{-1/2} \big) \\
&= \log \det((\Lambda_h + \lambda \mathcal{I}_\mathcal{H})) + \log \big( 1 + \phi(z)^\top (\Lambda_h + \lambda \mathcal{I}_\mathcal{H})^{-1} \phi(z) \big).
\end{aligned}
$$

Since $\phi(z)^\top (\Lambda_h + \lambda \mathcal{I}_\mathcal{H})^{-1} \phi(z) \leq 1$ for $\lambda > 1$, we have

$$
\begin{aligned}
\phi(z)^\top \big( \Lambda_h + \lambda \mathcal{I}_\mathcal{H} \big)^{-1} \phi(z) &\leq 2 \log \big( 1 + \phi(z)^\top (\Lambda_h + \lambda \mathcal{I}_\mathcal{H})^{-1} \phi(z) \big) \\
&= 2 \log \det(\Lambda_h(z) + \lambda \mathcal{I}_\mathcal{H}) - 2 \log \det(\Lambda_h + \lambda \mathcal{I}_\mathcal{H}) \\
&= 2 \log \det(I + K_h(z)/\lambda) - 2 \log \det(I + K_h/\lambda), \qquad (49)
\end{aligned}
$$

recall that $\Gamma_h(s,a) = \beta \cdot \|\phi(s,a)\|_{(\Lambda_h + \lambda \mathcal{I}_\mathcal{H})^{-1}}$ by equation 47 and equation 48, we have

$$\Gamma_h(s,a) \leq \sqrt{2}\beta \cdot (\log(I + K_h(z)/\lambda) - \log(I + K_h/\lambda))^{1/2}, \qquad (50)$$

for all $(s,a) \in \mathcal{S} \times \mathcal{A}$, and by Theorem D.1, we have

$$
\begin{aligned}
\text{SubOpt}(\{\tilde{\pi}_h\}) &\leq \sum_{h=1}^H \mathbb{E}_{\pi^*}[\Gamma_h(s_h, a_h)] \\
&\leq \sum_{h=1}^H \sqrt{2}\beta \cdot \mathbb{E}_{\pi^*}\big[ \{\log \det(I + K_h(z_h)/\lambda) - \log \det(I + K_h/\lambda)\}^{1/2} \big] \\
&\leq \sum_{h=1}^H \sqrt{2}\beta \cdot \big\{ \mathbb{E}_{\pi^*}\big[ \log \det(I + K_h(z_h)/\lambda) - \log \det(I + K_h/\lambda) \big] \big\}^{1/2} \\
&= \sum_{h=1}^H \sqrt{2}\beta \cdot \big\{ \mathbb{E}_{\pi^*}\big[ \phi(s_h, a_h)^\top (\Phi_h^\top \Phi_h + \lambda \mathcal{I}_\mathcal{H})^{-1} \phi(s_h, a_h) \big] \big\}^{1/2} \\
&= \sum_{h=1}^H \sqrt{2}\beta \, \text{Tr}\big( (K_h + \lambda \mathcal{I}_\mathcal{H})^{-1} \Sigma_h^* \big)^{1/2},
\end{aligned}
$$

where the first inequality comes from Theorem D.1, the second from equation 50, the third inequality from the feature map representation in equation 47. By Lemma D.19 in Jin et al. (2021), with $\lambda$ specified in Theorem 5.3, we have

$$\sum_{h=1}^{H} \text{Tr}\left((K_h + \lambda \mathcal{I}_{\mathcal{H}})^{-1}\Sigma_h^*\right)^{1/2} \leq 4d_{\text{eff}}^{\text{pop}}$$

for eigenvalue decaying conditions defined in Assumption 5.1. Therefore, for any $\delta \in (0,1)$, we set $\beta$ and $\lambda$ as in Theorem 5.3, then we can guarantee that

$$\text{SubOpt}(\{\tilde{\pi}_h\}_{h\in[H]}) \leq \mathcal{O}\left(\beta \cdot d_{\text{eff}}^{\text{pop}}\right).$$

Recall that we define

$$\beta = \begin{cases} C'' \cdot H \cdot \left\{\sqrt{\lambda}R_r + d_{\text{eff}}^{\text{sample}} e^H |\mathcal{A}| \cdot \log(nR_rH/\delta)^{1/2+1/(2\mu)}\right\} & \mu\text{-finite spectrum,} \\ C'' \cdot H \cdot \left\{\sqrt{\lambda}R_r + d_{\text{eff}}^{\text{sample}} e^H |\mathcal{A}| \cdot \log(nR_rH/\delta)^{1/2+1/(2\mu)}\right\} & \mu\text{-exponential decay,} \\ C'' \cdot H \cdot \left\{\sqrt{\lambda}R_r + d_{\text{eff}}^{\text{sample}} e^H |\mathcal{A}| \cdot (nR_r)^{\kappa^*} \cdot \sqrt{\log(nR_rH/\delta)}\right\} & \mu\text{-polynomial decay,} \end{cases}$$

we therefore conclude the proof of Theorem 5.3.

### E.3 Proof for Lemma E.4

We prove Lemma E.4 by discussing the eigenvalue decaying conditions in Assumption 5.1 respectively.

**(i):$\mu$-finite spectrum.** In this case, since $1 + 1/n \in [1,2]$, by Lemma F.7, there exists some absolute constant $C$ that only depends on $d, \mu$ such that

$$G(n, 1+1/n) \leq C \cdot \mu \cdot \log n,$$

and by Lemma F.8, there exists an absolute constant $C'$ such that

$$\log N(\mathcal{Q}, \|\cdot\|_{\infty}, 1/n) \leq C' \cdot \mu \cdot [\log(nR_rH) + C_4],$$

Hence we could set $\beta = c \cdot H \cdot \left(\sqrt{\lambda}R_r + d_{\text{eff}}^{\text{sample}} e^H |\mathcal{A}| \cdot \sqrt{\mu \log(nR_rH/\delta)}\right)$ for some sufficiently large constant $c > 0$.

**(ii): $\mu$-exponential decay.** By Lemma F.7, there exists some absolute constant $C$ that only depends on $d, \gamma$ such that

$$G(n, 1+1/n) \leq C \cdot (\log n)^{1+1/\mu},$$

and by Lemma F.8, there exists an absolute constant $C'$ such that

$$\log N(\mathcal{Q}, \|\cdot\|_{\infty}, 1/n) \leq C' \cdot \log(nR_r)^{1+1/\mu},$$

We can thus choose $\beta = c \cdot H \cdot (\sqrt{\lambda}R_r + d_{\text{eff}}^{\text{sample}} e^H |\mathcal{A}| \cdot \log(nR_rH/\delta)^{1/2+1/(2\mu)})$ for some sufficiently large absolute constant $c > 0$ depending on $d, \mu, C_1, C_2$ and $C_\psi$.

**(iii): $\mu$-polynomial decay.** By Lemma F.7, there exists some absolute constant $C$ that only depends on $d, \mu$ such that

$$G(n, 1+1/n) \leq C \cdot n^{\frac{d+1}{\mu+d}} \cdot \log n,$$

and by Lemma F.8, there exists an absolute constant $C'$ such that

$$\log N(\mathcal{Q}, \|\cdot\|_{\infty}, 1/n) \leq C' \cdot (nR_r)^{2/[\mu \cdot (1-2\tau)-1]} \cdot \log(nR_r),$$

Thus, it suffices to choose $\beta = c \cdot H \cdot (\sqrt{\lambda}R_r + d_{\text{eff}}^{\text{sample}} e^H |\mathcal{A}| \cdot (nR_r)^{\kappa^*} \cdot \sqrt{\log(nR_rH/\delta)})$, where $c > 0$ is a sufficiently large absolute constant depending on $d, \mu$. Here

$$\kappa^* = \frac{d+1}{2(\mu+d)} + \frac{1}{\mu(1-2\tau)-1}.$$

## F    AUXILIARY LEMMA

The following lemma is useful in the proof of Lemma F.2.

**Lemma F.1.** *For three symmetrical matrices $A, B$ and $C$, suppose $A \succeq B$ and $C \succeq 0$, we have*

$$\langle A, C \rangle \geq \langle B, C \rangle.$$

*Proof.* Consider

$$\langle A - B, C \rangle = \text{tr}\big((A - B)C\big).$$

Note that since $C$ is positive definite, we have a real symmetrical matrix $H$ such that $C = H^2$. Therefore we have

$$\text{tr}\big((A - B)C\big) = \text{tr}\big(H(A - B)H\big).$$

Denote $H$ by $(h_1, \cdots, h_d)$, we then have

$$\text{tr}\big(H(A - B)H\big) = \sum_{i=1}^{d} h_i^\top (A - B)h_i,$$

and by $A - B$ being semi-definite positive we conclude the proof.    □

The following lemma is useful when upper bounding the self-normalizing sequence.

**Lemma F.2.** *For real numbers $x_1, x_2, ..., x_n$ and real vectors $c_1, c_2, ..., c_n \in \mathcal{H}$, where $\mathcal{H}$ is a Hilbert space. If $\sum_{i=1}^{n} x_i^2 \leq C$, where $C > 0$ is a positive constant, then*

$$(\sum_{i=1}^{n} x_i c_i)(\sum_{i=1}^{n} x_i c_i)^\top \preceq C \cdot \sum_{i=1}^{n} c_i c_i^\top.$$

*Proof.* Consider an arbitrary vector $y \in \mathcal{H}$. We have

$$\begin{aligned}
y^\top (\sum_{i=1}^{n} x_i c_i)(\sum_{i=1}^{n} x_i c_i)^\top y &= \left\| \sum_{i=1}^{n} x_i \cdot (c_i \cdot y) \right\|_{\mathcal{H}}^2 \\
&\leq \left( \sum_{i=1}^{n} x_i^2 \right)\left( \sum_{i=1}^{n} (c_i \cdot y)^2 \right) \\
&\leq C \cdot \left( y^\top \sum_{i=1}^{n} c_i c_i^\top y \right),
\end{aligned}$$

since this holds for all $y \in \mathcal{H}$ we conclude the proof.

□

The following lemma upperly bounds the bracketling number of a parametrized function class by the covering number of the paramter class when it is Lipschitz-continuous to the parameter.

**Lemma F.3.** *Consider a class $\mathcal{F}$ of functions $m_\theta : \theta \in \Theta$ indexed by a parameter $\theta$ in an arbitrary index set $\Theta$ with a metric $d$. Suppose that the dependence on $\theta$ is Lipschitz in the sense that*

$$|m_{\theta_1}(x) - m_{\theta_2}(x)| \leq d(\theta_1, \theta_2) F(x)$$

*for some function $F : \mathcal{X} \to \mathbb{R}$, for every $\theta_1, \theta_2 \in \Theta$ and $x \in \mathcal{X}$. Then, for any norm $\| \cdot \|$, the bracketing numbers of this class are bounded by the covering numbers:*

$$N_{[]}(\mathcal{F}, \| \cdot \|, 2\epsilon \|F\|) \leq N(\Theta, d, \epsilon).$$

*Proof.* See Lemma 2.14 in Sen (2018) for details.    □

The following two lemmas, obtained from Abbasi-Yadkori et al. (2011), establishes the concentration of self-normalized processes.

**Lemma F.4.** *[Concentration of Self-Normalized Processes, (Abbasi-Yadkori et al., 2011)] Let $\{\epsilon_t\}_{t=1}^{\infty}$ be a real-valued stochastic process that is adaptive to a filtration $\{\mathcal{F}_t\}_{t=0}^{\infty}$. That is, $\epsilon_t$ is $\mathcal{F}_t$-measurable for all $t \geq 1$. Moreover, we assume that, for any $t \geq 1$, conditioning on $\mathcal{F}_{t-1}, \epsilon_t$ is a zero-mean and $\sigma$-subGaussian random variable such that*

$$\mathbb{E}\left[\epsilon_t \mid \mathcal{F}_{t-1}\right] = 0 \quad and \quad \mathbb{E}\left[\exp\left(\lambda \epsilon_t\right) \mid \mathcal{F}_{t-1}\right] \leq \exp\left(\lambda^2 \sigma^2/2\right), \quad \forall \lambda \in \mathbb{R}.$$

*Besides, let $\{\phi_t\}_{t=1}^{\infty}$ be an $\mathbb{R}^d$-valued stochastic process such that $\phi_t$ is $\mathcal{F}_{t-1}$-measurable for all $t \geq 1$. Let $M_0 \in \mathbb{R}^{d \times d}$ be a deterministic and positive-definite matrix, and we define $M_t = M_0 + \sum_{s=1}^{t} \phi_s \phi_s^{\top}$ for all $t \geq 1$. Then for any $\delta > 0$, with probability at least $1 - \delta$, we have for all $t \geq 1$ that*

$$\left\|\sum_{s=1}^{t} \phi_s \cdot \epsilon_s\right\|_{M_t^{-1}}^2 \leq 2\sigma^2 \cdot \log\left(\frac{\det\left(M_t\right)^{1/2} \det\left(M_0\right)^{-1/2}}{\delta}\right).$$

*Proof.* See Theorem 1 of Abbasi-Yadkori et al. (2011) for detailed proof. □

**Lemma F.5** (Concentration of Self-Normalized Process for RKHS, (Chowdhury & Gopalan, 2017)). *Let $\mathcal{H}$ be an RKHS defined over $\mathcal{X} \subseteq \mathbb{R}^d$ with kernel function $K(\cdot, \cdot) : \mathcal{X} \times \mathcal{X} \to \mathbb{R}$. Let $\{x_\tau\}_{\tau=1}^{\infty} \subset \mathcal{X}$ be a discrete-time stochastic process that is adapted to the filtration $\{\mathcal{F}_t\}_{t=0}^{\infty}$. Let $\{\epsilon_\tau\}_{\tau=1}^{\infty}$ be a real-valued stochastic process such that (i) $\epsilon_\tau \in \mathcal{F}_\tau$ and (ii) $\epsilon_\tau$ is zero-mean and $\sigma$-sub-Gaussian conditioning on $\mathcal{F}_{\tau-1}$, i.e.,*

$$\mathbb{E}\left[\epsilon_\tau \mid \mathcal{F}_{\tau-1}\right] = 0, \quad \mathbb{E}\left[e^{\lambda \epsilon_\tau} \mid \mathcal{F}_{\tau-1}\right] \leq e^{\lambda^2 \sigma^2/2}, \quad \forall \lambda \in \mathbb{R}$$

*Moreover, for any $t \geq 2$, let $E_t = (\epsilon_1, \ldots, \epsilon_{t-1})^{\top} \in \mathbb{R}^{t-1}$ and $K_t \in \mathbb{R}^{(t-1) \times (t-1)}$ be the Gram matrix of $\{x_\tau\}_{\tau \in [t-1]}$. Then for any $\eta > 0$ and any $\delta \in (0, 1)$, with probability at least $1 - \delta$, it holds simultaneously for all $t \geq 1$ that*

$$E_t \left[\left(K_t + \eta \cdot I\right)^{-1} + I\right]^{-1} E_t \leq \sigma^2 \cdot \log \det\left[(1 + \eta) \cdot I + K_t\right] + 2\sigma^2 \cdot \log(1/\delta)$$

*Proof.* See Theorem 1 in Chowdhury & Gopalan (2017) for detailed proof. □

The following theorem gives a uniform bound for a set of self-normalizing sequences, whose proof can be found in Appendix B.2, Jin et al. (2021). It is useful for uniformly bounding the self-normalizing sequence in pessimistic value iteration, both for linear model MDP class:

**Theorem F.6.** *For $h \in [H]$, we define the function class $\mathcal{V}_h(R, B, \lambda) = \{V_h(x; \theta, \beta, \Sigma) : \mathcal{S} \to [0, H] \text{ with } \|\theta\| \leq R, \beta \in [0, B], \Sigma \succeq \lambda \cdot I\}$, where*

$$V_h(x; \theta, \beta, \Sigma) = \max_{a \in \mathcal{A}} \left\{\min\left\{\phi(x, a)^{\top}\theta - \beta \cdot \sqrt{\phi(x, a)^{\top}\Sigma^{-1}\phi(x, a)}, H - h + 1\right\}_+\right\},$$

*then we have*

$$\sup_{V \in \mathcal{V}_{h+1}(R, B, \lambda)} \left\|\sum_{i=1}^{n} \phi\left(s_h^i, a_h^i\right) \cdot \left(V(s_{h+1}^i) - \mathbb{E}[V(s_{h+1}) \mid s_h^i, a_h^i]\right)\right\|_{(\Lambda_h + \lambda I)^{-1}}^2$$
$$\leq 8\epsilon^2 n^2/\lambda + 2H^2 \cdot \left(2 \cdot \log(\mathcal{N}/\delta) + d \cdot \log(1 + n/\lambda)\right),$$

*holds with probability at least $1 - \delta$ for every $\epsilon > 0$. Here*

$$\log(\mathcal{N}) \leq d \cdot \log(1 + 4R/\varepsilon) + d^2 \cdot \log(1 + 8d^{1/2}B^2/(\varepsilon^2\lambda)).$$

*Proof.* See Appendix B.2 in Jin et al. (2021) for details. □

**Lemma F.7** (Lemma D.5 in Yang et al. (2020)). *Let $\mathcal{Z}$ be a compact subset of $\mathbb{R}^d$ and $K : \mathcal{Z} \times \mathcal{Z} \to \mathbb{R}$ be the RKHS kernel of $\mathcal{H}$. We assume that $K$ is a bounded kernel in the sense that $\sup_{z \in \mathcal{Z}} K(z, z) \leq 1$, and $K$ is continuously differentiable on $\mathcal{Z} \times \mathcal{Z}$. Moreover, let $T_K$ be the integral operator induced by $K$ and the Lebesgue measure on $\mathcal{Z}$, whose definition is given in equation 9. Let $\{\sigma_j\}_{j \geq 1}$ be the eigenvalues of $T_K$ in the descending order. We assume that $\{\sigma_j\}_{j \geq 1}$*

*satisfy either one of the following three eigenvalue decay conditions: (i) $\mu$-finite spectrum: We have $\sigma_j = 0$ for all $j \geq \mu + 1$, where $\mu$ is a positive integer. (ii) $\mu$-exponential eigenvalue decay: There exist constants $C_1, C_2 > 0$ such that $\sigma_j \leq C_1 \exp\left(-C_2 \cdot j^\mu\right)$ for all $j \geq 1$, where $\mu > 0$ is positive constant. (iii) $\mu$-polynomial eigenvalue decay: There exists a constant $C_1$ such that $\sigma_j \geq C_1 \cdot j^{-\mu}$ for all $j \geq 1$, where $\mu \geq 2 + 1/d$ is a constant.*

*Let $\sigma$ be bounded in interval $[c_1, c_2]$ with $c_1$ and $c_2$ being absolute constants. Then, for conditions (i)-(iii) respectively, we have*

$$G(n, \lambda) \leq \begin{cases} C_n \cdot \mu \cdot \log n & \mu\text{-finite spectrum,} \\ C_n \cdot (\log n)^{1+1/\mu} & \mu\text{-exponential decay,} \\ C_n \cdot n^{(d+1)/(\mu+d)} \cdot \log n & \mu\text{-polynomial decay,} \end{cases}$$

*where $C_n$ is an absolute constant that depends on $d, \mu, C_1, C_2, C, c_1,$ and $c_2$.*

*Proof.* See Lemma D.5 in Yang et al. (2020) for details. $\qquad \square$

**Lemma F.8** ( $\ell_\infty$-norm covering number of RKHS ball). *For any $\epsilon \in (0, 1)$, we let $N(\mathcal{Q}, \|\cdot\|_\infty, \epsilon)$ denote the $\epsilon$-covering number of the RKHS norm ball $\mathcal{Q} = \{f \in \mathcal{H} : \|f\|_\mathcal{H} \leq R\}$ with respect to the $\ell_\infty$-norm. Consider the three eigenvalue decay conditions given in Assumption 5.1. Then, under Assumption 5.1, there exist absolute constants $C_3$ and $C_4$ such that*

$$\log N(\mathcal{Q}, \|\cdot\|_\infty, \epsilon) \leq \begin{cases} C_3 \cdot \mu \cdot [\log(R/\epsilon) + C_4] & \mu\text{-finite spectrum,} \\ C_3 \cdot [\log(R/\epsilon) + C_4]^{1+1/\mu} & \mu\text{-exponential decay,} \\ C_3 \cdot (R/\epsilon)^{2/[\mu \cdot (1-2\tau)-1]} \cdot [\log(R/\epsilon) + C_4] & \mu\text{-polynomial decay,} \end{cases}$$

*where $C_3$ and $C_4$ are independent of $n, H, R,$ and $\epsilon$, and only depend on absolute constants $C_\psi$, $C_1, C_2, \mu,$ and $\tau$ specified in Assumption 5.1.*

*Proof.* See Lemma D.2 in Yang et al. (2020) for details. $\qquad \square$

