# OpenReview forum: "Reinforcement Learning with Human Feedback: Learning Dynamic Choices via Pessimism"
_ICLR.cc/2024/Conference — ICLR 2024 Conference Withdrawn Submission_

### Official Review · Reviewer_CQn7 · 2023-10-28

**Soundness:** 3 good
**Presentation:** 3 good
**Contribution:** 2 fair
**Rating:** 6
**Confidence:** 3

**Summary:**

This paper studies Reinforcement Learning under Human Feedback (RLHF) setting where human feedback is not myopic. They model this as an offline MDP setting and tackle the three main challenges unique to RLHF (which is similar to offline RL): (1) The agent must infer human behavior policies from the offline data. (ii) The agent must tackle a dynamic environment and estimate a reward function from behavior policies. (iii) Finally tackle the challenge of insufficient dataset coverage and large state space. To do this they propose a dynamic choice pessimistic policy optimization algorithm. This is a value iteration-based algorithm that incorporates a penalty term (pessimism) that ensures learning in an offline setting (standard for offline RL setting). Thy study three settings and establish sub-optimality bounds for them. These are the settings for the general model class, linear model class and RKHS model class and all of them suffer from an additional $e^H$ term. This is a theory RL paper and they do not conduct any experiments.

**Strengths:**

1) Understanding the RLHF setting and its connection to offline RL is an important direction of research.
2) They introduce a standard algorithm with the penalty term (pessimism) to tackle the three challenges of RLHF.
3) They theoretically analyze three settings and establish sub-optimality bounds for them.

**Weaknesses:**

1) Some assumptions need more justification.
2) The learning of the human preference model is not clear.
3) Discussions on some factors in the sub-optimality gaps are missing.
4) The connection to offline learning is clear but the difference and key technical novelty needs to be discussed more.
5) No discussions on lower bound.

**Questions:**

1) One of my main concerns is understanding the Assumption 3.2 (and subsequent similar assumption) on model identification. Is this $a_0$ similar to a safe action as in conservative/constraint bandits (or MDP)? Why is this required and where doe this show up?
2) Another concern is the missing discussion on technical novelty. This paper does not discuss where does their proof differs from Jin et al (say) for the linear MDP setting. Can you elaborate on this? Similar papers on theoretical RL has extensive discussion on their main proof. (see Uehara orr Jin et al.)
3) Another concern is the extra factor of $e^H$ in the bound. Where and how do they show up? Is it really necessary or is it possible to get rid of them?
4) A detailed discussion on the main difference between RLHF an offline MDP settings is missing. Your proofs seem to leverage many of their techniques. Can you elaborate on the main similarities/differences of the proof technique briefly? This will help me put te paper in context.
5) Finally please discuss your idea on the lower bound (at least for the linear MDP setting which is well understood). This will help us understand if the $e^H$ factor is really necessary.

**Details Of Ethics Concerns:**

Not applicable.

---

> ### Author Response · Authors · 2023-11-15
> **Thank you for your review!**
>
> Dear Reviewer CQn7,
>
> Thank you for your detailed review! In the following, we will try to address your concerns.
>
> Q1. One of my main concerns is understanding Assumption 3.2 (and subsequent similar assumptions) on model identification. Is this similar to a safe action as in conservative/constraint bandits (or MDP)? Why is this required and where does this show up?
>
> A1. Assumption 3.2 aims to tackle the issue of identification: we cannot compare reward $Q(s_1,a)$ and $Q(s_2,a')$ without such an assumption when $s_1 \neq s_2$, since both $Q(s,a) + A(s)$ would results in the same behavior policy according to Equation (1) for arbitrary function $A(s)$. Without such an assumption, the underlying $Q^{\pi_b}$ could be unidentifiable even if we know the human behavior. For convenience, in this paper we also assume the  $f\in\mathcal{M}_h$ satisfies $f(s,a_0)=0$. However, in practice, we can simply modify by setting $\hat{Q}(s,a) = \hat{Q}(s,a) - \hat{Q}(s,a_0)$ to achieve the same result.
>
> We also claim that our results also hold under other identification assumptions. For example, [1] assumes that $\theta_h^\top \cdot 1 = 0$.
>
> Q2. Another concern is the missing discussion on technical novelty. This paper does not discuss where does their proof differs from Jin et al (say) for the linear MDP setting. Can you elaborate on this? Similar papers on theoretical RL have an extensive discussion on their main proof. (see Uehara orr Jin et al.)
>
> A2. We confirm with the reviewer that in the pessimistic value iteration part, our method is akin to Jin et al.. However, we illustrate by their method to show that our results can be easily tailored to existing methods in offline RL, as we can successfully bound the uncertainty quantification of our reward estimation. We would like to character our theoretical contribution in two aspects:
>
> (1) We tackle the issue of how to learn reward from a non-myopic human agent in a dynamic choice model, which is an important subject for modeling human preference in an MDP ([7][8][9]). Current literatures such as ([1][2][4])  use trajectory-based preference to learn $\sum_{h=1}^Hr_h(s_h,a_h)$. To precisely recover single-step reward $r_h(s,a)$ from trajectory cumulative reward, these methods further need the reward to be homogenous, e.g. $r_h = r$ ([1],[2],[3],[8]) or $r_h = \phi_h\cdot \theta$ for some fixed $\theta$ ([4]).  However, when such assumption does not hold, e.g. consider the simple case of (say) $r_h(s,a) = \phi(s,a)\cdot\theta_h$, we cannot exactly recover reward $r_h(s,a)$ from $\sum_{h=1}r(s_h,a_h)$ for every $h\in[H]$, which could lead to suboptimality in the learned policy. In this paper, we tackle this issue by (a) first learning the human behavior policy and her value function, then (b) recovering underlying utility by minimizing Bellman-MSE. To our best knowledge, such a framework is novel.
>
> (2) We provide theoretical proof that the reward learned in our method can be easily tailored to existent offline reinforcement learning algorithms by characterizing the uncertainty quantification $\Gamma_h$. Specifically, we show that the uncertainty quantification is of order $O(e^H \|\phi\|_{\Lambda_h^{-1}})$, which is almost the same order as simple ridge regression when the reward is known. We claim that such an analysis is non-trivial since we are using a **two-stage** algorithm, which is hard to analyze in many cases, and usually requires doubly robust methods[10]. Specifically, we illustrate the efficiency of our method by tailoring reward learning to pessimistic value iteration, which is widely studied in theoretical RL ([4][5][6]), and show that our algorithm achieves a $O(1/\sqrt{T})$ suboptimality with high probability under a standard coverage assumption. To the best of our knowledge, this is the first provably efficient model for learning from human feedback under a dynamic choice model.

---

> ### Author Response · Authors · 2023-11-15
> **Thank you for your review!**
>
> Q3. Another concern is the extra factor of $e^H$ in the bound. Where and how do they show up? Is it really necessary or is it possible to get rid of them? Please discuss your idea on the lower bound (at least for the linear MDP setting which is well understood). This will help us understand if the $e^H$ factor is really necessary.
>
> A3. The term $e^H$ shows up due to reversing the learned human policy to Q-function: with parametric MLE, we can guarantee $\mathbb{E}[TV(\pi(\cdot|s), \hat{\pi}(\cdot|s))^2]\leq O(1/N)$. Since $\pi(a|s) = \exp(Q(s,a))/\sum_{a}\exp(Q(s,a))$ and $Q\leq H$, this results in the $O(|\mathcal{A}|e^H/N)$ convergence rate in MSE for $Q$ function. In this paper we assume $r_h\in[0,1]$, and for general case when $\sum_{h=1}^H r_h(s_h,a_h) \leq B$, such a factor would become $e^B$.
>
> For the current paper studying learning reward from human preference or risk-sensitive RL, which both pertain to learning reward from logistic regression, how to eliminate the exponential dependency is in general unknown even for easier settings such as pairwise trajectory-based preference, see Theorem 3.2, Theorem 4.1 in [1] and Theorem 1 in [2],  which we illustrate as a special case in Appendix A.
>
> Results for lower bound exists for RL under trajectory-based comparison setting also exists, e.g. see Theorem 3.10 in [1]. However, such a bound takes of form of $c\cdot \sqrt{1/N}$, where $c$ only concerns coverabily and dimension, therefore doesn't match the exponential factor in the upper bound, and therefore cannot prove/disprove the necessity of the exponential term.
>
> From an asymptotic point of view, consider the simplified case of $|\mathcal{A}| = 2$,  under the assumption of $Q = \phi\cdot \theta$,  the famous Cramer-Rao lower bound shows that the variance matrix of our estimate $\hat{\theta}$ is lower bounded by  $\mathbb{E}_{s\sim \pi_b}[\Sigma]^{-1}$, where $\Sigma$ is defined by $$
> \frac{\exp(Q(s,a_0)+Q(s,a_1))}{(\exp(Q(s,a_0))+\exp(Q(s,a_1)))^2}(\phi(s,a_0) - \phi(s,a_1))(\phi(s,a_0) - \phi(s,a_1))^\top,
> $$
> In the worse case when $Q(s,a_0) \equiv 0$, $Q(s,a_1)\equiv H$, this could result in an $\exp(H)/\sqrt{n}$ factor in the asymptotic error of $\hat{\theta}$, and would also show up in the estimation error of $r_h$ if we use Bellman MSE minimization in the following procedure.
>
> Q4. A detailed discussion on the main difference between RLHF and offline MDP settings is missing. Your proofs seem to leverage many of their techniques. Can you elaborate on the main similarities/differences of the proof technique briefly? This will help me put te paper in context.
>
> A4. We appreciate the reviewer for bringing this to our attention. **Offline RL** tackles the issue of learning the optimal policy in MDP from an offline dataset, which contains state-action transitions and reward observation. In offline RL, the main difficulty lies in how to deal with intrinsic uncertainty and spurious correlation. Uehara et al. and Jin et al. show that with a method called pessimism value iteration, the suboptimality can be bounded by $\mathbb{E}[\sum_{h=1}^H\Gamma_h]$, where $\Gamma_h$ is the uncertainty quantifier.  In **RLHF, or preference-based RL**, the reward is not explicitly observed. We need to learn the reward function from human behavior and utilize the learned reward to learn the optimal policy([6][7][8][1]). In such cases, the technical difficulty mainly lies in how to learn the reward, and how to characterize the estimation error.
>
> In this paper, we tailor methods from both fields: (1)First, we learn the reward function from a human-generated trajectory dataset, which generalizes trajectory-based preference methods([1][2][3]). Such a case is more difficult than trajectory-based preference, since the human makes decision at every state $s_h$, and her decision takes future randomness into account. Moreover, we consider an inhomogenous reward, $r_h = \phi\cdot\theta_h$, where $\theta_h$ is different for every $h\in[H]$. Therefore, simple one-step logistic regression in [1][9] is insufficient. Our methods include (i) learning human policy and corresponding value function by maximum likelihood estimation, and (ii) learning the reward by minimizing a Bellman MSE. Such a technique is novel to our best knowledge. (2)Next, we characterize the uncertainty quantification in the reward estimation process theoretically. Surprisingly, the uncertainty quantification of reward is almost of the same order as the case the reward is observable. We further show that by delicately choosing $\Gamma_h$,  we can achieve a $O(1/\sqrt{T})$ suboptimality with high probability.

---

> ### Author Response · Authors · 2023-11-15
> **References**
>
> >[1]B. Zhu, J. Jiao, and M. I. Jordan. Principled reinforcement learning with human feedback from pairwise or k-wise comparisons, 2023.
>
> >[2]Zhan, Wenhao, et al. "Provable Offline Reinforcement Learning with Human Feedback." arXiv preprint arXiv:2305.14816 (2023).
>
> >[3]Zhan, Wenhao, et al. "How to Query Human Feedback Efficiently in RL?." arXiv preprint arXiv:2305.18505 (2023).
>
> >[4]Fei, Yingjie, et al. "Exponential bellman equation and improved regret bounds for risk-sensitive reinforcement learning." Advances in Neural Information Processing Systems 34 (2021): 20436-20446.
>
> >[5]Yingjie Fei, Zhuoran Yang, and Zhaoran Wang. Risk-sensitive reinforcement learning with function approximation: A debiasing approach. In International Conference on Machine Learning, pages 3198–3207. PMLR, 2021.
>
> >[6] Wirth C, Akrour R, Neumann G, Fürnkranz J. A survey of preference-based reinforcement learning methods. Journal of Machine Learning Research. 2017 Dec 1;18(136):1-46.
>
> >[7]Ouyang L, Wu J, Jiang X, et al. Training language models to follow instructions with human feedback[J]. Advances in Neural Information Processing Systems, 2022, 35: 27730-27744.
>
> >[8]Lee, Kimin, Laura Smith, and Pieter Abbeel. "Pebble: Feedback-efficient interactive reinforcement learning via relabeling experience and unsupervised pre-training." arXiv preprint arXiv:2106.05091 (2021).]
>
> >[9]Ziebart, Brian D., et al. “Maximum entropy inverse reinforcement learning.” Aaai. Vol. 8. 2008.
>
> >[10]Rothe, Christoph, and Sergio Firpo. "Properties of doubly robust estimators when nuisance functions are estimated nonparametrically." Econometric Theory 35.5 (2019): 1048-1087.

---

> > ### Comment · Reviewer_CQn7 · 2023-11-21
> > **Response to Rebuttal**
> >
> > Thanks for your response. I have no further questions. I am positive about this work.

---

### Official Review · Reviewer_Ejar · 2023-11-09

**Soundness:** 3 good
**Presentation:** 3 good
**Contribution:** 3 good
**Rating:** 5
**Confidence:** 3

**Summary:**

The main objective of this paper is to investigate the realm of offline Reinforcement Learning in scenarios where the reward is not directly observable. To address this challenge, the authors introduce the DCPPO algorithm. To elaborate, the authors start by making assumptions about the agent's policy based on certain characteristics and then proceed to recover the concealed reward function through human feedback. Once the reward is estimated, the DCPPO algorithm incorporates it into a standard RL framework to discover a policy that is close to optimal. Ultimately, the authors provide a theoretical guarantee regarding the sub-optimality of the resulting policy.

**Strengths:**

1. In contrast to conventional RL algorithms, this study does not depend on an observable reward function. Rather, it endeavors to recover the concealed reward through human feedback.

2. The paper is skillfully composed and presents information in a comprehensible manner.

**Weaknesses:**

1. This study lacks novelty as the algorithm is essentially a straightforward integration of a reward learning framework with a standard reinforcement learning algorithm, offering little in terms of groundbreaking innovation.

2.The assumption regarding the agent's policy in equation (1) is overly restrictive. Specifically, it mandates that every agent has access to the value function $Q_h^{\pi}$, which may be unattainable if the agent lacks prior knowledge of the transition probability function $P_h$. If the transition probability function $P_h$ is already known, Algorithm 2 becomes redundant, and it suffices to focus solely on estimating the reward as in Algorithm 1.

3. The estimation error discussed in equation (9) may encounter issues stemming from data dependency. To elucidate, the estimated reward is constructed in Algorithm 1 and relies on the offline dataset. However, in Algorithm 2 (Line 2), the agent performs ridge regression using the same dataset. Due to this data dependency, it is possible that $E[\tilde V (s_{h+1})]\ne P_h \tilde V (s_h,a_h)$. A similar issue has been addressed in the work by Jin et al. (2019) through the application of uniform convergence techniques. Unfortunately, in this study, the estimation reward function lacks a linear structure, and the corresponding value function class may not have a small covering number. Consequently, even with the use of uniform convergence techniques, the problem may not be resolved.

[1] Provably Efficient Reinforcement Learning with Linear Function Approximation

**Questions:**

See Weaknesses.

---

> ### Author Response · Authors · 2023-11-15
> **Thank you for your review!**
>
> Dear reviewer Ejar,
>
> Thank you for your review! In the following, we will try to address your concerns.
>
> Q1.This study lacks novelty as the algorithm is essentially a straightforward integration of a reward learning framework with a standard reinforcement learning algorithm, offering little in terms of groundbreaking innovation.
>
> A1. We would like to summarize our contribution as follows:
>
> (1)We tackle the issue of how to learn reward from a non-myopic human agent in a dynamic choice model, which is an important subject for modeling human preference in an MDP ([7][8][9]). Existing methods, relying on human preference between trajectories, learn reward of a whole trajectory, i.e. $\sum_{h\in[H]} r_h$ from single-step logistic regression. To recover the single-step reward $r_h(s,a)$, they need to assume **(1)** the reward is homogenous, i.e. $r_h(s,a) = r(s,a)$ for all $h$ ([2],[3]),  or **(2)** the reward model satisfies $r_h(s,a) = \phi_h(s,a)\cdot \theta$ for some fixed vector $\theta$([1][4]).
>
> However, existing methods cannot recover the single-step reward $r_h$ when a homogenous reward fails to hold, for example, consider the simple case of $r_h = \phi\cdot \theta_h$ for some feature $\phi$, we cannot directly recover all $\theta_h$ from trajectory-based preference due to **inhomogeneity**. Under such circumstances, existing methods **cannot learn the optimal policy**. In this paper, we tackle this issue by **(1)** first learning the human behavior policy and their value function by MLE, and **(2)** recovering underlying utility by minimizing Bellman-MSE. To our best knowledge, such a framework is novel;
>
> (2) We give a theoretical guarantee that the reward learned by our method can be easily tailored to the existence offline reinforcement learning algorithm. Specifically, we characterize the uncertainty quantification $\Gamma_h$ for reward function $r_h$. Such analysis is non-trivial since we are using a **two-stage algorithm**, which is known to be hard to analyze in general, and usually requires doubly robust methods[18]. We further illustrate that by simple pessimistic value iteration, our algorithm achieves a $O(1/\sqrt{T})$ suboptimality with high probability. To our knowledge, this is the first paper that provides a theoretical guarantee for learning optimal policy with human feedback made under a dynamic choice model.
>
> Q2.The assumption regarding the agent's policy in equation (1) is overly restrictive. Specifically, it mandates that every agent has access to the value function, which may be unattainable if the agent lacks prior knowledge of the transition probability function.  If the transition probability function is already known, Algorithm 2 becomes redundant, and it suffices to focus solely on estimating the reward as in Algorithm 1.
>
> A2. Thanks for bringing up this confusing point. However, we would like to claim that such an assumption is widely accepted in various areas. Equation (1) assumes that the agent takes action according to the **dynamic choice model** ([7]), which is a generalization of the famous Bradley-Terry-Luce model([4]) used in fine-tuning of language model ([4][11]) and MaxEnt IRL ([1]), and has been widely accepted to model human choices in various fields([7][8][9][11][13][14]). Such an assumption is also **weaker** than the Boltzman policy in Bayesian inverse RL ([15][16][17]), in which the agent takes action according to the softmax policy of the optimal Q-function. We would also like to humbly point out that **we do not require that the underlying transition is known to us**, and need to (1) infer the reward function from the behavior dataset and (2) learn the optimal policy from the inferred reward and dataset. Therefore Algorithm 2 is not redundant. We promise to clarify this point in the future version of our work.

---

> ### Author Response · Authors · 2023-11-15
> **Thank you for your review!**
>
> Q3.The estimation error discussed in equation (9) may encounter issues stemming from data dependency. To elucidate, the estimated reward is constructed in Algorithm 1 and relies on the offline dataset. However, in Algorithm 2 (Line 2), the agent performs ridge regression using the same dataset. Due to this data dependency, it is possible that $E[\hat{V}(s_{h+1})]\neq P_h \hat{V}(s_h,a_h)$. A similar issue has been addressed in the work by Jin et al. (2019) through the application of uniform convergence techniques. Unfortunately, in this study, the estimation reward function lacks a linear structure, and the corresponding value function class may not have a small covering number. Consequently, even with the use of uniform convergence techniques, the problem may not be resolved. Due to this data dependency, it is possible that ${E}[\hat{V}(s_{h+1})]!={P}_h \hat{V}(s_h,a_h)$. A similar issue has been addressed in the work by Jin et al. (2019)
>
> A3. We appreciate that you bring up this potentially confusing point; however, we humbly disagree with your statement. First, we confirm with the reviewer that for the linear case, we are using a uniform law of large number (ULLM) to control the weighted 2-norm of a zero mean empirical process (Lemma F.6), see page 21. To address your concern, in this work **we are using a linear function/RKHS approximation for the estimated reward function**, see Section 3.2 and Equation 5 in our main paper. The covering numbers of function classes can be bounded whenever $\|\hat{w}_h+\tilde{u}_h\|_2$ is bounded. Since we prove this in Lemma D.2, our proof with ULLM makes sense.
> >[1] Ziebart, Brian D., et al. “Maximum entropy inverse reinforcement learning.” Aaai. Vol. 8. 2008.
>
> >[2]Zhou, Yang, Rui Fu, and Chang Wang. “Learning the car-following behavior of drivers using maximum entropy deep inverse reinforcement learning.” Journal of Advanced Transportation 2020 (2020): 1-13.
>
> >[3]Wulfmeier, Markus, Peter Ondruska, and Ingmar Posner. “Maximum entropy deep inverse reinforcement learning.” arXiv preprint arXiv:1507.04888 (2015).
>
> >[4]B. Zhu, J. Jiao, and M. I. Jordan. Principled reinforcement learning with human feedback from pairwise or k-wise comparisons, 2023.
>
> >[5]Jin, Ying, Zhuoran Yang, and Zhaoran Wang. "Is pessimism provably efficient for offline rl?." International Conference on Machine Learning. PMLR, 2021.]
>
> >[6]Uehara, Masatoshi, and Wen Sun. "Pessimistic model-based offline reinforcement learning under partial coverage." arXiv preprint arXiv:2107.06226 (2021).
>
> >[7]V. Aguirregabiria and P. Mira. Dynamic discrete choice structural models: A survey. Journal of Econometrics, 156(1):38–67, 2010.
>
> >[8] Practical Methods for Estimation of Dynamic Discrete Choice Models, Annual Review of Economics, Peter Arcidiacono and Paul B. Ellickson
>
> >[9]K. Adusumilli and D. Eckardt. Temporal-difference estimation of dynamic discrete choice models. arXiv preprint arXiv:1912.09509, 2019.
>
> >[10]Zhan, Wenhao, et al. “Provable Offline Reinforcement Learning with Human Feedback.” arXiv preprint arXiv:2305.14816 (2023).
>
> >[11]Ouyang, Long, et al. "Training language models to follow instructions with human feedback." Advances in Neural Information Processing Systems 35 (2022): 27730-27744.]
>
> >[12]Zeng, Siliang, et al. "Maximum-likelihood inverse reinforcement learning with finite-time guarantees." Advances in Neural Information Processing Systems 35 (2022): 10122-10135.
>
> >[13]Zeng, Siliang, Mingyi Hong, and Alfredo Garcia. "Structural Estimation of Markov Decision Processes in High-Dimensional State Space with Finite-Time Guarantees." arXiv preprint arXiv:2210.01282 (2022).
>
> >[14]Sharma, Mohit, Kris M. Kitani, and Joachim Groeger. "Inverse reinforcement learning with conditional choice probabilities." arXiv preprint arXiv:1709.07597 (2017).
>
> >[15]Arora, Saurabh, and Prashant Doshi. "A survey of inverse reinforcement learning: Challenges, methods and progress." Artificial Intelligence 297 (2021): 103500.
>
> >[16]Choi, Jaedeug, and Kee-Eung Kim. "Nonparametric Bayesian inverse reinforcement learning for multiple reward functions." Advances in neural information processing systems 25 (2012).

---

> > ### Comment · Reviewer_Ejar · 2023-11-15
> >
> > I appreciate the author's clarification regarding my third question. I had some misunderstanding about the estimated reward function, but now I believe there is no issue with the estimation error discussed in equation (9). However, my concern about the agent's policy in equation (1) from question 2 persists. While it is true that a similar assumption (Bradley-Terry-Luce model) has been employed in previous reinforcement learning literature [1], these models typically base their distribution on the reward for the trajectory. It appears more reasonable that the agent knows the reward function or, at the very least, possesses some preferences for the final trajectory. In contrast, the current work assumes the agent's policy follows the dynamic choice model with respect to the value function, incorporating bounded-rationality denoted as $Q_h^{\pi_b}$. This assumption may seem overly restrictive. Additionally, it would be more reasonable if the policy depended on the optimal policy rather than the bounded-rational value function, although this is still more restrictive than relying solely on the reward. Overall, I will increase my score to 5.
> >
> > [1] B. Zhu, J. Jiao, and M. I. Jordan. Principled reinforcement learning with human feedback from pairwise or k-wise comparisons

---

> > > ### Author Response · Authors · 2023-11-20
> > > **Any further opinions?**
> > >
> > > Dear reviewer,
> > >
> > > As the deadline for the reviewing session approaches, we would like to confirm if we have addressed all of your concerns.

---

> > > > ### Comment · Reviewer_Ejar · 2023-11-21
> > > >
> > > > Thanks for the authors respondse and my concern from question 2 persists. Thus, I will keep my score.

---

> ### Author Response · Authors · 2023-11-17
> **Thank you for your comment!**
>
> We appreciate the reviewer's effort for bringing up this interesting question to our attention, and would like to further address your concern on dynamic discrete choice model. We believe that establishing our setting under the current model (rather than the BTL model or Boltzman based IRL) offers the following benefits:
>
> **(i)** When modeling sequential human behaviors in a dynamic environment, the current dynamic choice model is among the most common practice, and has been one of **the most widely-studied choice models in the past several decades**, both from a theoretical and empirical percepective in statistics and economics. For a comprehensive view, we would like to invite the reviewer the check ([1][2][3][4]). In such a setting, we do not require the human agent to gain a full understanding of **the probability transition**, but only require her to decide her action with respect to the soft-max policy of her own Q-value function,e.g. see Equation (13) in [1], where the value function $v(x,a)$ corresponds to our state-action value function, and shares the same formulation as equation (1). Such an setting has also been testified in pratice, e.g. the empirical results in [2][3][6] and other related works in our paper.
>
> **(ii)** We claim that the policy defined by $\pi_b\propto \exp(Q^{\pi_b})$ is also of significance in inverse RL. Such a definition corresponds to the K.K.T condition of maximum **causal entropy** inverse RL (MCE IRL), in which the sampled trajectory aims to maximize the causal entropy $H(a_{1:H}||s_{1:H})$ while matching feature expectation, see Lemma 3.2 in [5]; while the trajectory-based comparison BTL model aims to maximize the **Shannon entropy** $H(s_{1:H}, a_{1:H})$, e.g. see Remark 2.1 of [5]. The same remark also proves that directly using Shannon entropy to calculates the entropy over the entire trajectory distribution could cause an unwanted dependency on transition entropy, which further cause bias. Under such a logic, MCE IRL is more reasonable when learning from an inverse RL. Since equation (1) in our paper corresponds to MCE IRL, our method is reasonbale in the context of inverse RL.
>
> **(iii)** As we previously claimed in A2, directly learning from a trajectory-based comparison could be inefficient when the underlying reward function suffers from intrinsic inhomogenous, and could therefore results in suboptimality in policy learning.
>
> **(iv)** We humbly disagree with the reviewer by claiming that  **knowing $Q^{\pi_*}$** is actually more restrictive than **knowing $Q^{\pi_b}$**, since $\pi_b$ is the human's self policy, and accessing knowledge of her own policy is more reasonable than knowing the global optimal policy $\pi^*$. knowledge of $\pi^*$ would require access of the optimal Q-function $Q^*$, which could be highly impractical in practice.
>
> In the future version of our work, we plan to include the discussions above in the Introduction/Related Work section.
>
>
>
> >[1]Aguirregabiria, Victor, and Pedro Mira. "Dynamic discrete choice structural models: A survey." Journal of Econometrics 156.1 (2010): 38-67.
>
> >[2]Imai, Susumu, Neelam Jain, and Andrew Ching. "Bayesian estimation of dynamic discrete choice models." Econometrica 77.6 (2009): 1865-1899.
>
> >[3]Bajari, Patrick, et al. Identification and efficient semiparametric estimation of a dynamic discrete game. No. w21125. National Bureau of Economic Research, 2015.
>
> >[4]Chernozhukov, Victor, et al. "Locally robust semiparametric estimation." Econometrica 90.4 (2022): 1501-1535.
>
> >[5]Gleave, Adam, and Sam Toyer. "A primer on maximum causal entropy inverse reinforcement learning." arXiv preprint arXiv:2203.11409 (2022).
>
> >[6] Van der Klaauw, Wilbert. "On the use of expectations data in estimating structural dynamic choice models." Journal of Labor Economics 30.3 (2012): 521-554.

---

> ### Author Response · Authors · 2023-11-21
> **We care for your concern**
>
> Dear reviewer,
>
> Could you elaborate more on your concern about our previous answer? We would like to summarize our previous answer of **why dynamic choice model defined in equation (1) is important and makes sense**:
>
> **(i)** Dynamic choice model defined in equation (1), as an important setting to capture human preference under dynamic environment, has been widely studied both from theoretical and empirical aspects, and is of interest to multiple subjects;
>
> **(ii)** The policy defined in equation 1 corresponds to the **Maximum Causal Entropy IRL**, which has been proven to be a better setting from the aspect of learning human behavior than the **Maximum Shannon Entropy IRL**, the latter corresponds to the trajectory-based human preference. We also recommend the reviewer to a detailed literature;
>
> **(iii)** Dynamic choice model defined in equation (1) can tackle the issue of reward inhomogeneity in the underlying setting, while trajectory-based comparison cannot, as we demonstrated in the previous A2;
>
> **(iv)** In comparison to the case where the human policy is a softmax policy with respect to $Q^{\pi^*}$, we claim that the optimal policy $\pi^*$ could be much more difficult to obtain than the human's self-behavior policy $\pi_b$, and therefore our setting is more reasonable.
>
> We kindly want to bring to the reviewer's attention that we have provided details on these arguments in our previous response. We are keen to better understand the reason behind any remaining concerns you have, and hope to provide a more detailed explanation to ensure all related concerns are addressed!

---

### Official Review · Reviewer_e24G · 2023-11-09

**Soundness:** 2 fair
**Presentation:** 3 good
**Contribution:** 2 fair
**Rating:** 5
**Confidence:** 3

**Summary:**

This paper handles the case of Reinforcement Learning from Human Feedback (RLHF) where the underlying reward function is not available and the optimal policy needs to be estimated from human state-action trajectories. This is a setting similar to Inverse RL where the reward function is learned from human demonstrations to train the agent via RL loop. The paper uses Dynamic-Choice-Pessimistic-Policy-Optimization to find the optimal policy under the special case of a dynamically evolving MDP, for example, the RLHF case where the reward is obtained at each intermediate step of token generation as opposed to the final token.

**Strengths:**

The paper looks at a special case of optimal policy learning under dynamic choice where the underlying MDP seems to be changing after each choice made which is not considered in the usual RLHF literature. The paper proposes an unique algorithms Dynamic-Choice-Pessimistic-Policy-Optimization (DCPPO) to handle this situation.

**Weaknesses:**

**Dynamic choice setting**: The paper claims that it handles the special cases of (a) unobserved rewards that should be learned from the human trajectory, and (b) dynamic choice where the underlying MDP is dynamically evolving. For the first case, the settings seem similar to generalized Max entropy Inverse RL works (which is addressed in the next point). Secondly, the paper gives an example of the dynamic rewards for RLHF, where rewards are obtained at intermediate steps as well. However, I am not sure why this is referred to as `dynamic nature of MDP transition` because this is similar to MDP settings, assuming state $s_t = x_{1:t}$ based on which the probability of the next token will be affected. This can be handled with standard offline algorithms, and therefore I am not clear on the motivation behind DCPPO.

**Similarity to Max Ent Inverse RL**: As mentioned above, this work seems similar to Max Ent Inverse RL works. The paper mentioned in the appendix that this work is a generalization of such previous works, however, I am still not convinced that MaxEnt IRL method would not be applicable to the mentioned setting because of the above reason. Additionally, there has been some work to generalize MaxEnt IRL works [1, 2]. Given this, I am not sure why Dynamic Discrete Choice might be necessary for RLHF settings and if they are practical.

**Empirical results**: The paper does not provide any empirical analysis of DCPPO compared to other online/offline algorithms. Even a simple comparison would have clarified the importance of DCPPO in such settings and would also help with the above issues.

**Questions:**

Questions to the author:
- Could you clarify the setting under which DCPPO would be necessary and practically useful and why existing IRL methods cannot handle such settings?
- In the paper, is there a special consideration for language model training because the term RLHF seems to indicate that? If not, is this work addressing a special setting for inverse RL/imitation learning?

---

> ### Author Response · Authors · 2023-11-15
> **Thank you for your review!**
>
> Dear Reviewer e24G,
>
> Thank you for your reivew! In the following we would try to address your concerns.
>
> Q1. Could you clarify the setting under which DCPPO would be necessary and practically useful and why existing IRL methods cannot handle such settings?
>
> A1. This is a good point. Existing IRL methods,  such as ([1][2][5]) and RLHF in InstructGPT, learn the reward function by a trajectory-based comparison, i.e. the loss function is $r= \log(\exp(\sum_{h=1}^H r(s_h^i,a_h^i))/Z(r))$, where $Z(r) = \sum_{\tau}\exp(\sum_{h=1}^H r(s_h^i,a_h^i))$. Such a method could be **suboptial** under the following settings:
>
> (1) When the agent acts in a look-ahead manner---the agent cannot directly prefer a whole trajectory, but can only decide her action $a_h$ in every $s_h$, which incorporates randomness caused by the environment in future steps. Such a data-generating process models human's non-myopic nature has already been widely accepted by economics (e.g. [9], also see the Related Work in our paper), and is more complicated than simply choosing a trajectory from all possible trajectories. Under such a setting, employing the maxEnt IRL method alone suffers model misspecification, and leads to significant suboptimality.
>
> (2) Existing preference-based IRL methods cannot learn an inhomogeneous reward. Existing IRL works ([1],[2],[3],[4]) either assumes a homogenous reward i.e. $r_h = r$ for all $h\in[H]$, or assumes a linear reward $r_h = \phi_h\cdot\theta$ with the same feature vector $\theta$. Such methods then infers $\sum_{h} r_h(s_h,a_h)$ from trajectory-based preference, and recovers the single step $r_h$ by the learned $r$ or $\theta$. However, these methods are restrictive when the ground-truth reward is not homogenous. For example, even when the linear reward takes a form $r_h = \phi\cdot \theta_h$, we cannot even recover the single step reward $r_h(s,a)$ from the learned cumulative reward $\sum_{h=1}^H r_h(s_h,a_h)$ by simple MaxEnt IRL, and therefore cannot guarantee to learn the optimal policy.
>
> In our paper, we tackle such difficulty by a two-stage algorithm: (a) we use logistic regression to estimate Q-function, (b) we recover reward function from Q-function by minimizing Bellman MSE. Furthermore, we show the efficiency of our method by tailoring it to pessimistic value iteration and show that our method can attain a $O(1/\sqrt{T})$ with only single policy coverage. While existing IRL methods **cannot** cover our setting as shown by (1) and (2) , **our method can cover the case of MaxEnt IRL and RLHF in InstructGPT**, and we recommend the reviewer to see Appendix A for details.
>
> Q2. In the paper, is there a special consideration for language model training because the term RLHF seems to indicate that? If not, is this work addressing a special setting for inverse RL/imitation learning?
>
> A2. Thanks for bringing this to our attention! In this paper, we adopt **a broader view of RLHF** — the RL agent does not have access to the reward function, and the reward needs to be inferred from human preferences. Such a view is widely adopted before OpenAI popularized the notion of RLHF. See, for example, the reference [6]. Under this broader view, human feedback can possibly take many forms:
> **(i)** showing the RL agent an optimal action at the current state — this is the setting considered by DAGGER in [7];
> **(ii)** Labeling each state-action by a label “preferable” or “not” — this is our version and we illustrate that it covers **(iii)** and **(iv)** in Appendix A. In our model, P(a is preferable given s) $\propto \exp(Q^{\pi_b}(s,a))$. This view is also adopted in many existing IRL works (prior to OpenAI’s InstructGPT paper), e.g., [8], and a large amount of economics literature, see [9];
> **(iii)** Ranking trajectory pairs by preference — $P(traj_1 > traj_2 ) \propto \exp( \sum_{h} r_h(s_h, a_h) - \sum_h r_h(\tilde s_h, \tilde a_h))$. This is the view adopted by InstructGPT;
> **(iv)** Trajectories are labeled as "good" or "bad." The probability that a trajectory is labeled as good is proportional to $\exp\left(\sum_{h} r_h(s_h, a_h)\right)$. This viewpoint aligns with the approach in Max Ent IRL.
>
> All the above, in this paper, we aim to learn a more general framework of RLHF than fine-tuning language models/ MaxEnt IRL from a theoretical point of view. We promise to elaborate on this view in our Introduction/Related Works section in the future version of our work.

---

> > ### Author Response · Authors · 2023-11-15
> > **References**
> >
> > >[1] Ziebart, Brian D., et al. "Maximum entropy inverse reinforcement learning." Aaai. Vol. 8. 2008.
> >
> > >[2]Zhou, Yang, Rui Fu, and Chang Wang. "Learning the car-following behavior of drivers using maximum entropy deep inverse reinforcement learning." Journal of Advanced Transportation 2020 (2020): 1-13.
> >
> > >[3]Wulfmeier, Markus, Peter Ondruska, and Ingmar Posner. "Maximum entropy deep inverse reinforcement learning." arXiv preprint arXiv:1507.04888 (2015).
> >
> > >[4]B. Zhu, J. Jiao, and M. I. Jordan. Principled reinforcement learning with human feedback from pairwise or k-wise comparisons, 2023.
> >
> > >[6] Wirth C, Akrour R, Neumann G, Fürnkranz J. A survey of preference-based reinforcement learning methods. Journal of Machine Learning Research. 2017 Dec 1;18(136):1-46.
> >
> > > [7] Ross, S., Gordon, G. and Bagnell, D., 2011, June. A reduction of imitation learning and structured prediction to no-regret online learning. In Proceedings of the fourteenth international conference on artificial intelligence and statistics (pp. 627-635). JMLR Workshop and Conference Proceedings.
> >
> > >[8] Griffith S, Subramanian K, Scholz J, Isbell CL, Thomaz AL. Policy shaping: Integrating human feedback with reinforcement learning. Advances in neural information processing systems. 2013;26.
> >
> > >[9] V. Aguirregabiria and P. Mira. Dynamic discrete choice structural models: A survey. Journal of Econometrics, 156(1):38–67, 2010.

---

> ### Author Response · Authors · 2023-11-20
> **Any further opinions?**
>
> Dear reviewer,
>
> As the deadline for the reviewing session approaches, we would like to confirm if we have addressed all of your concerns.